# Breaking AR's Sampling Bottleneck: Provable Acceleration via Diffusion Language Models

**Gen Li**[*]
Chinese University of Hong Kong
genli@cuhk.edu.hk

**Changxiao Cai**[*†]
University of Michigan
cxcai@umich.edu

## Abstract

Diffusion models have emerged as a powerful paradigm for modern generative modeling, demonstrating strong potential for large language models (LLMs). Unlike conventional autoregressive (AR) models that generate tokens sequentially, diffusion models allow for parallel sampling, offering a promising path to accelerate generation and eliminate the left-to-right generation constraints. Despite their empirical success, theoretical understandings of diffusion language models remain underdeveloped. In this work, we develop convergence guarantees for diffusion language models from an information-theoretic perspective. Our analysis demonstrates that the sampling error, measured by the Kullback-Leibler (KL) divergence, decays inversely with the number of iterations $T$ and scales linearly with the mutual information between tokens in the target text sequence. Crucially, our theory covers the regime $T < L$, where $L$ is the text sequence length. This justifies that high-quality samples can be generated with fewer iterations than $L$, thereby breaking the fundamental sampling bottleneck of $L$ steps required by AR models. We further establish matching upper and lower bounds, up to some constant factor, that shows the tightness of our convergence analysis. These results offer novel theoretical insights into the practical effectiveness of diffusion language models.

## 1 Introduction

Large language models (LLMs) fall within the domain of generative modeling, which aim to learn the unknown probability distribution of natural language from training data. The state-of-the-art LLMs are typically trained using an autoregressive (AR) modeling paradigm. For a text sequence of $L$ tokens $x = (x^{(1)}, \ldots, x^{(L)})$, an AR model factorizes the joint distribution as

$$p(x) = p(x^{(1)}) \prod_{i=2}^{L} p(x^{(i)} \mid x^{(1)}, \ldots, x^{(i-1)}), \tag{1}$$

and generate tokens sequentially from left to right. Despite its remarkable success (Radford et al., 2018, 2019; Brown et al., 2020), the AR approach suffers from several notable drawbacks. First, token generation is constrained by a rigid left-to-right order, prohibiting the model from reasoning earlier tokens based on later context. Second, the one-by-one generation is inherently slow, as tokens are produced one at a time, limiting the efficiency of sampling.

Motivated by the above limitations and the extraordinary performance of diffusion models in various generative modeling tasks (Sohl-Dickstein et al., 2015; Song and Ermon, 2019; Ho et al., 2020; Song et al., 2020), recent research has begun exploring diffusion models as an alternative approach to

---

[*]The authors contribute equally.

[†]Corresponding author.

39th Conference on Neural Information Processing Systems (NeurIPS 2025).

language modeling (Dieleman et al., 2022; Han et al., 2022; Gulrajani and Hashimoto, 2023; He et al., 2022). Unlike the AR paradigm, diffusion language models allow parallel sampling of tokens through an iterative denoising process, thereby eliminating left-to-right constraints and potentially accelerating text generation. Discrete diffusion models have emerged as a promising framework for LLMs in this vein (Austin et al., 2021; Campbell et al., 2022; Lou et al., 2023), which is tailored to generate discrete-structured samples.

Among the discrete diffusion models, one notable class is the masked diffusion model (Austin et al., 2021; Shi et al., 2024; Sahoo et al., 2024). It introduces an absorbing state called *mask* and achieves the best empirical performance. Identical to its continuous counterpart, the masked diffusion model consists of two complementary processes: a forward process that progressively corrupts a text sequence $X_0 \sim p_{\text{data}}$ drawn from the data distribution by masking out tokens:

$$X_0 \stackrel{\text{mask}}{\to} X_1 \stackrel{\text{mask}}{\to} X_2 \stackrel{\text{mask}}{\to} \cdots \stackrel{\text{mask}}{\to} X_T;$$

a reverse process that learns to reconstruct the original sequence by iteratively predicting the masked tokens:

$$Y_0 \stackrel{\text{unmask}}{\leftarrow} Y_1 \stackrel{\text{unmask}}{\leftarrow} Y_2 \stackrel{\text{unmask}}{\leftarrow} \cdots \stackrel{\text{unmask}}{\leftarrow} Y_T.$$

The mask predictors—conditional distributions that take partially masked sequences as input and predict masked tokens—serve a role analogous to the score estimators in continuous diffusion models, guiding the reverse process to recover the text.

Compared to the AR paradigm, diffusion modeling offers several key advantages for language generation:

- *Sampling acceleration.* By generating multiple tokens in parallel at each iteration, diffusion models can reduce the number of sampling iterations and speed up the overall sampling process compared to one-token-at-a-time AR generation[3].

- *Reversal reasoning.* Without a unidirectional order, diffusion language models can perform reverse generation tasks (for example, inferring earlier tokens from later ones) that are impossible for standard AR models constrained to a forward-only generation.

- *Controllable generation.* Because diffusion models do not follow a strictly left-to-right generation order, they can more easily incorporate global constraints or planning for long-range dependencies, enabling more flexible control over the generated text (Li et al., 2022).

These benefits have spurred a surge of interest in diffusion language models. A flurry of recent works has demonstrated the viability of diffusion models for language models, showing that they can achieve comparable performance to AR approaches in certain settings (Lou et al., 2023; Sahoo et al., 2024; Gong et al., 2024; Campbell et al., 2024; Nie et al., 2025; Ye et al., 2023). Moreover, diffusion language models have been shown to handle generation tasks beyond the reach of AR methods, such as reversal reasoning, which standard AR models cannot perform (Nie et al., 2025).

However, despite their empirical promise, rigorous theory for diffusion language models remains in its infancy. In particular, there is limited insights into how the quality of the generated text relates to the sampling procedure or to the statistical structure of the underlying language distribution. Only until very recently have researchers begun to explore its sampling guarantees. The work (Chen and Ying, 2024) examines convergence guarantees of discrete diffusion models in terms of total variation (TV) distance and Kullback-Leibler (KL) divergence. However, their analysis is restricted to regimes where, on average, *less than one token* is masked per step. This assumption does not align with practical diffusion language models that mask a large fraction of tokens at each iteration (Yu et al., 2025). Such a gap between practice and theory motivates the central question of our study:

*Given accurate mask predictors, can we establish the convergence guarantees of diffusion language models for general sampling procedures and data distribution?*

---

[3]While diffusion language models enable parallel sampling, current practical implementations are typically slower than highly optimized AR models with KV caching (Pope et al., 2023). Recent work demonstrates that distillation can close part of this gap (Deschenaux and Gulcehre, 2024; Hayakawa et al., 2024); see Sahoo et al. (2024, Fig. 2) for a comparison of text generation speed.

**Main contributions.** In light of the above gap, this paper takes an initial step towards a convergence theory for diffusion language models from an information-theoretic perspective. We seek to rigorously characterize the quality of the generated samples (i.e., sampling error) as a function of the number of iterations steps and the statistical structure of target text distribution.

To make the analysis tractable, we adopt a standard decoupling approach in prior theoretical analyses of diffusion models (Block et al., 2020; De Bortoli et al., 2021; Chen et al., 2022a, 2023a; Li et al., 2024; Li and Yan, 2024; Li and Cai, 2024; Li et al., 2025), which separates the training stage (how to learn the mask predictors) and the sampling phase (how to generate samples). Our work focuses on the latter, assuming access to a given mask predictor and analyzing the sampling procedure.

Under this setup, we establish the first convergence guarantees of diffusion language models for general sampling schemes and data distributions. In particular, our analysis shows that after $T$ iterations, the KL divergence between the output distribution and the true data distribution decays on the order of $1/T$, with a coefficient governed by the information coupling among tokens. Specifically, we prove an upper bound on the sampling error (measured by the KL divergence) of the form:

$$O\left(\frac{1}{T}\sum_{i=1}^{L} I(X^{(i)}; X^{(-i)})\right) + \varepsilon_{\mathsf{train}},$$

where $I(X^{(i)}; X^{(-i)})$ denotes the mutual information between the $i$-th token $X^{(i)}$ and the remaining tokens $X^{(-i)}$ under the data distribution $X \sim p_{\mathsf{data}}$, and $\varepsilon_{\mathsf{train}}$ captures the training error due to imperfect mask predictors (see Section 2 for a formal definition). Notably, our theory accommodates the regime where the number of iterations $T$ is smaller than the sequence length $L$, which provides a formal justification for the sampling acceleration of diffusion language models over their AR counterparts. Further, we complement this upper bound with a matching lower bound (up to constant factors), showing that our convergence analysis is tight. In other words, the $1/T$ decay of error and its linear dependence on the sequence's mutual information cannot be substantially improved in general.

Our theoretical findings, grounded in information theory, provide new insights into why diffusion language models can be so effective in practice. The above guarantee holds for a broad class of data distributions, suggesting that diffusion language models have robust performance across diverse language data. Moreover, by linking convergence to the mutual information among tokens, our results highlight how the statistical dependencies in language data influence the efficiency of parallel diffusion sampling.

## 1.1 Other related work

**Discrete diffusion models.** While diffusion models were initially introduced for both discrete and continuous state spaces in the seminal work (Sohl-Dickstein et al., 2015), subsequent studies have predominantly focused on Gaussian diffusion processes in continuous domains. Applying diffusion models to intrinsically discrete settings is challenging because Gaussian noise cannot be directly applied to corrupt discrete-valued data. Prior works on discrete diffusion models can be broadly categorized into two classes. The first class embeds discrete structures into a continuous space and applies continuous diffusion (Chen et al., 2022b; Dieleman et al., 2022; Gulrajani and Hashimoto, 2023; Han et al., 2022; Li et al., 2022; Lovelace et al., 2023; Strudel et al., 2022). The second class directly defines the forward process on discrete structures using various categorical Markov transition matrices (Hoogeboom et al., 2021; Austin et al., 2021; Sahoo et al., 2024), often under the continuous-time Markov chain (CTMC) framework. This perspective has further led to methods for adapting score matching (Song and Ermon, 2019) to discrete settings (Meng et al., 2022; Sun et al., 2022; Lou et al., 2023).

**Theory for diffusion models.** Our work is closely related to the convergence theories for continuous diffusion models in $\mathbb{R}^d$—a field that is considerably more mature than its discrete counterpart. These studies address a fundamental question: given imperfect score estimates, how many iterations are required to sample accurately from the target distribution? Under the assumption of $L^2$-accurate score estimates and a log-Sobolev inequality for the target distribution, Lee et al. (2022) established the first polynomial iteration complexity bounds. Later works relaxed these assumptions by either imposing Lipschitz continuity on the scores (Chen et al., 2022a; Lee et al., 2023) or by requiring bounded support/moment conditions for the target distribution (Chen et al., 2023a). The current state-of-the-art results, as derived in Benton et al. (2023) and Li and Yan (2024), achieve convergence rate of

$\widetilde{O}(\sqrt{d/T})$ in KL divergence and $\widetilde{O}(d/T)$ in total variation distance, respectively. In addition to the convergence analysis, recent work has established end-to-end statistical guarantees by characterizing the errors in the score estimation and sampling stage. These analyses yield rigorous bounds on the sampling error in diverse distributional settings, such as smooth densities (Oko et al., 2023; Chen et al., 2023b; Wibisono et al., 2024; Zhang et al., 2024; Dou et al., 2024; Cai and Li, 2025) and Gaussian mixture models (Gatmiry et al., 2024; Chen et al., 2024).

## 1.2 Notation

For integer $n > 0$, we denote $[n] \coloneqq \{1, 2, \ldots, n\}$. For $x > 0$, we use $\lceil x \rceil$ to denote the smallest integer greater than or equal to $x$ and $\lfloor x \rfloor$ to denote the largest integer less than or equal to $x$. Let $\mathbb{X}$ denote the (discrete) vocabulary of texts. We use $\mathsf{M}$ to denote the mask and extend the vocabulary $\mathbb{X}$ by including a single point $\{\mathsf{M}\}$ to obtain $\overline{\mathbb{X}} = \mathbb{X} \cup \{\mathsf{M}\}$. For vector $x \in \mathbb{X}^L$, we use $x^{(i)}$ to represent its $i$-th entry for $i \in [L]$. Moreover, for any set $M \subset [L]$, we use $x \circ M = (x_i)_{i \in M}$ to denote the vector in $\mathbb{X}^{|M|}$ that consists of the entries of $x$ indexed by the set $M$. In addition, let $\mathcal{P}_M : \mathbb{X}^L \to \overline{\mathbb{X}}^L$ denote the projection defined as

$$[\mathcal{P}_M(x)]_i = \begin{cases} x_i, & i \in M, \\ \mathsf{M}, & i \notin M. \end{cases} \tag{2}$$

For a random variable $X$, we use $p_X$ to denote its distribution and probability density function interchangeably for simplicity of notation. For random vectors $(X, Y) \sim p_{X,Y}$ with marginal distributions $p_X$ and $p_Y$, let $\mathsf{KL}(p_X \,\|\, p_Y) \coloneqq \int p_X(x) \log \frac{p_X(x)}{p_Y(x)} \, \mathrm{d}x$ denote the Kulback-Leibler divergence between $p_X$ and $p_Y$. The mutual information between $X$ and $Y$ is defined as $I(X; Y) \coloneqq \mathsf{KL}(p_{X,Y} \,\|\, p_X p_Y)$. For random vectors $(X, Y, Z) \sim p_{X,Y,Z}$, the conditional mutual information between $X$ and $Y$ given $Z$ is defined as $I(X; Y \mid Z) \coloneqq \mathsf{KL}(p_{XY|Z} p_Z \,\|\, p_{X|Z} p_{Y|Z} p_Z)$.

For two functions $f(n), g(n) > 0$, we use $f(n) \lesssim g(n)$ or $f(n) = O\big(g(n)\big)$ to mean $f(n) \leq Cg(n)$ for some absolute constant $C > 0$. Similarly, we write $f(n) \gtrsim g(n)$ or $f(n) = \Omega\big(g(n)\big)$ when $f(n) \geq C'g(n)$ for some absolute constant $C' > 0$. We denote $f(n) \asymp g(n)$ or $f(n) = \Theta\big(g(n)\big)$ when $Cf(n) \leq g(n) \leq C'f(n)$ for some absolute constants $C' > C > 0$.

## 2 Preliminaries

In this section, we provide a brief introduction to diffusion language models.

**Forward process.** Consider a text sequence $X_0 \in \mathbb{X}^L$ of length $L$ drawn from the data distribution $p_{\mathsf{data}}$. The forward process gradually corrupts $X_0$ by masking its tokens step by step until reaching a fully masked sequence $(\mathsf{M}, \ldots, \mathsf{M}) \in \overline{\mathbb{X}}^L$. In more detail, let $\{s_t\}_{t=1}^T$ be a sequence of positive integers such that $\sum_{t=1}^T s_t = L$. We call it mask size schedule since it defines how many tokens to mask at each step. We then construct a sequence of increasing mask index sets $\varnothing = M_0 \subseteq M_1 \subseteq \cdots \subseteq M_T = [L]$, where each $M_t$ is obtained by adding $s_t$ new indices chosen uniformly at random from the previously unmasked positions $M_{t-1}^c$. Formally, at each step $t \in [T]$, we select a subset $M_t \setminus M_{t-1}$ of $s_t$ token positions from $M_{t-1}^c$ uniformly at random and mask those positions, and let $M_t$ denote the set of all masked positions at step $t$. We denote by $X_t$ the partially masked sequence at step $t$, obtained from the original $X_0$ by replacing tokens at the masked positions $M_t$ with the mask symbol $\mathsf{M}$. Using the projection operator $\mathcal{P}_{M_t^c}$ defined in (2), we can write the sequence at step $t$ as

$$X_t = \mathcal{P}_{M_t^c}(X_0), \tag{3}$$

meaning $X_t$ retains the original tokens in positions not in $M_t$ and has $\mathsf{M}$ in positions $M_t$. After $T$ steps, $X_T = (\mathsf{M}, \ldots, \mathsf{M}) \in \overline{\mathbb{X}}^L$ is the fully masked sequence.

**Training.** The reverse process aims to invert the forward masking: starting from the fully masked sequence, it iteratively unmasks tokens to recover a sample from $p_{\mathsf{data}}$. The core of the diffusion language model is a mask predictor $p(\cdot \mid X_t)$ that represents the conditional distribution of the masked tokens given the partially observed sequence $X_t$. To learn the mark predictor, we fit the

generative model to the data distribution by minimizing a variational upper bound on the negative log-likelihood.

As directly modeling the joint distribution of all masked tokens can be intractable in high dimensions, practitioners typically parametrize the mask predictor using a factorized form:

$$p(x \mid X_t) = \prod_{i=1}^{L} p_i(x^{(i)} \mid X_t), \tag{4}$$

i.e., each token is predicted independently given $X_t$. We then seek a product distribution $p = \prod_{i=1}^{L} p_i$ that solves the following minimization problem:

$$\min_{p=\prod_{i=1}^{L} p_i} -\mathbb{E}_{\tau, X_0, M_\tau}\left[ \frac{L}{|M_\tau|} \sum_{i \in M_\tau} \log p_i(X_0^{(i)} \mid X_\tau) \right], \tag{5}$$

where the expectation is taken over a random time $\tau \in [T]$ with $\mathbb{P}\{\tau = t\} = s_t/L$ for $t \in [T]$, a training sample $X_0 \sim p_{\mathsf{data}}$ draw from the data distribution, and a random mask set of size $|M_\tau|$ chosen uniformly at random from $[L]$. Notice that the loss in (5) is computed over masked tokens. In practice the objective in (5) is approximated by its empirical average over the finite training samples.

As a remark, let $p^\star = \prod_{i=1}^{L} p_i^\star$ denote the optimal predictor (i.e., the minimizer of (5)). Then one can verify that for each $i \in [L]$, $p_i^\star(\cdot \mid X_t)$ coincides with the true conditional distribution $p_{X_0^{(i)} \mid X_t}(\cdot \mid X_t)$ of the $i$-token $X_0^{(i)}$ given the partially masked sequence $X_t$.

**Sampling procedure.** Once the mask predictor $\widehat{p}$ is trained, we generate new text by simulating the reverse process. Initializing at step $T$ with $M_T = [L]$ and $Y_T = (\mathsf{M}, \ldots, \mathsf{M}) \in \overline{\mathbb{X}}^L$, we iterate for $t = T, T-1, \ldots, 1$ as follows. We first choose a subset of $s_t$ masked positions to reveal, consistent with the forward schedule. Formally, we sample a mask set $M_{t-1} \subseteq M_t$ such that $M_t \setminus M_{t-1}$ consists of $s_t$ indices chosen uniformly at random from $M_t$ (the currently masked positions). Next, we sample placeholder values for the tokens in $M_t \setminus M_{t-1}$ using the learned mask predictor $\widehat{p}$ and current iterate $Y_t$:

$$Y_{t-1} := \mathcal{P}_{M_t^c}(Y_t) + \mathcal{P}_{M_t \setminus M_{t-1}}(\widehat{X}_t) \quad \text{with} \quad \widehat{X}_t \sim \widehat{p}(\cdot \mid Y_t). \tag{6}$$

Equivalently, we sample each masked position $i \in M_t \setminus M_{t-1}$ from $\widehat{p}_i(\cdot \mid Y_t)$ and leave the already unmasked positions $i \notin M_t$ as they are in $Y_t$. We then fill in those sampled tokens to obtain the next sequence $Y_{t-1}$, while keeping other positions fixed. After repeating this procedure down to $t = 1$, we output a fully unmasked sequence $Y_0 \in \mathbb{X}^L$.

## 3 Main results

In this section, we present the convergence guarantees for the sampling procedure of diffusion language models (see (6)).

To begin with, we introduce the following definition to characterize the quality of the mask predictor $\widehat{p}$ used in the sampling process.

**Definition 1.** For a mask predictor estimator $\widehat{p} = \prod_{i=1}^{T} \widehat{p}_i$, define its training error as

$$\varepsilon_{\mathsf{train}} := \mathbb{E}_{\tau, X_0, M_\tau}\left[ \frac{L}{|M_\tau|} \sum_{i \in M_\tau} \log p_i^\star(X_0^{(i)} \mid X_\tau) \right] - \mathbb{E}_{\tau, X_0, M_\tau}\left[ \frac{L}{|M_\tau|} \sum_{i \in M_\tau} \log \widehat{p}_i(X_0^{(i)} \mid X_\tau) \right], \tag{7}$$

where $p^\star$ is the minimizer of the objective (5).

In essence, the training error $\varepsilon_{\mathsf{train}}$ measures the likelihood gap caused by imperfect training of the mask predictor.

### 3.1 Sampling error upper bound

With the above definition, we now state our main results. We first present the sampling error upper bound. The proof is deferred to Section 4.

**Theorem 1.** *For any mask size schedule $\{s_t\}_{t=1}^T$, let $s_{\max} := \max_{t \in [T]} s_t$ be the maximum mask size. Also, let $M := (M_1, \ldots, M_T)$ denote the sequence of mask sets. Then the output $Y_0$ of the sampling procedure* (6) *satisfies*

$$\mathbb{E}_M \big[ \mathsf{KL}(p_{X_0} \| p_{Y_0 | M}) \big] \leq \frac{2^{\lceil \log_2 s_{\max} \rceil} - 1}{L} \sum_{i=1}^L I(X_0^{(i)}; X_0^{(-i)}) + \varepsilon_{\mathsf{train}}. \tag{8}$$

*Here, the expectation is taken over the randomness in the mask sets $M_1, \ldots, M_T$.*

Our result demonstrates that the sampling error—measured by the KL divergence between the output distribution $p_{Y_0}$ and the data distribution $p_{\mathsf{data}}$—consists of two components: an information-theoretic term depending on the data distribution $p_{\mathsf{data}}$ and an estimation term $\varepsilon_{\mathsf{train}}$ arising from imperfect mask predictions. It is noteworthy that the result holds for arbitrary mask size schedules $\{s_t\}_{t=1}^T$, which covers parallel sampling schemes where multiple tokens are unmasked per step ($s_t > 1$), and thus the number of iterations $T$ can be less than the sequence length $L$.

The first term captures the difficulty of modeling the token dependencies: it is the sum of mutual information between each token and the rest of the sequence $\sum_{i=1}^L I(X_0^{(i)}; X_0^{(-i)})$, scaled by a factor that depends on the mask size schedule $\{s_t\}_{t=1}^T$. The dependence on the mutual information quantifies how the intrinsic coupling of tokens in the data affects the difficulty of sampling while the second term $\varepsilon_{\mathsf{train}}$ reflects the training error of the mask predictor.

Notably, if the mask predictor is optimal (i.e., $\varepsilon_{\mathsf{train}} = 0$), then the sampling error is governed purely by the information structure of the data distribution. In general, the bound indicates that the more statistically dependent the sequence tokens are (higher mutual information), the larger the potential sampling error, unless more refined mask size schedules are used to compensate.

Furthermore, under a balanced mask size schedule where the mask sizes are set roughly uniform across iterations (i.e., $s_t \asymp L/T$ for all $t \in [T]$ and thus $s_{\max} \asymp L/T$), the leading term in Theorem 1 simplifies to $O(1/T)$ and we obtain a cleaner bound:

**Corollary 1.** *Suppose $\frac{1}{T} \sum_{t=1}^T s_t \asymp s_{\max}$. Then the output $Y_0$ of the sampling procedure* (6) *satisfies*

$$\mathbb{E}_M \big[ \mathsf{KL}(p_{X_0} \| p_{Y_0 | M}) \big] \leq \frac{C_1}{T} \sum_{i=1}^L I(X_0^{(i)}; X_0^{(-i)}) + \varepsilon_{\mathsf{train}} \tag{9}$$

*where $C_1 = T s_{\max} / \sum_{t=1}^T s_t \asymp 1$ is an absolute constant. Here, the expectation is taken over the randomness in the mask sets $M_1, \ldots, M_T$.*

In this regime, after $T$ iterations the sampling error becomes $O(1/T)$, with a prefactor given by the total mutual information $\sum_{i=1}^L I(X_0^{(i)}; X_0^{(-i)})$ of the sequence. In the idealized case $\varepsilon_{\mathsf{train}} = 0$, to achieve a target error level $\varepsilon$ in KL divergence, one needs on the order of $O(1/\varepsilon)$ iterations (up to a maximum of order $L$, since we cannot iterate more times than the sequence length without saturating the improvement). Meanwhile, if $\varepsilon_{\mathsf{train}}$ is nonzero, the final sampling error will decrease to a floor on the order of $\varepsilon_{\mathsf{train}}$. In other words, the sampling error increases proportionally to the training error, underscoring the importance of accurate mask prediction.

**Comparison with prior work.** The recent work by Feng et al. (2025) examines the efficiency of masked diffusion models for $n$-gram language model, where each token is generated based on its preceding $n - 1$ tokens (Brown et al., 1992). To quantify token-level accuracy, they introduce token error rate (TER), defined via perplexity:[4]

**Definition 2.** Given a data distribution $p_{X_0}$ and an output distribution $p_{Y_0}$, the TER is defined as

$$\log_2 \mathrm{TER}(p_{Y_0}; p_{X_0}) := -\frac{1}{L} \mathbb{E}_{X_0} \big[ \log p_{Y_0}(X_0) \big]. \tag{10}$$

---

[4]They also analyze the inefficiency of masked diffusion models via sequence error rate (SER), which falls beyond the scope of this paper.

When $n$ is a fixed constant (independent of the sequence length $L$), Feng et al. (2025) shows that a masked diffusion model can achieve a small TER using a few iterations, which is independent of sequence length $L$. However, their bound on TER scales as $\left((n-1)/T\right)^{1/n} \log |\mathbb{X}|$, which is suboptimal for any $n > 1$ and becomes increasingly loose as $n$ grows. Indeed, consider a trivial baseline that samples $Y_0 \sim p_0$ uniformly at random from all length-$L$ sequences, i.e., $p_0 \sim \mathsf{Unif}(\mathbb{X}^L)$. For this baseline, one can verify that $\log_2 \mathrm{TER}(p_0; p_{X_0}) - \log_2 \mathrm{TER}(p_{X_0}; p_{X_0}) \leq \log |\mathbb{X}|$. To beat this when $n \geq \log L$, the result of Feng et al. (2025) requires $T \gtrsim (n-1)4^n \gg L$, which is substantially larger than the sequence length $L$. Consequently, their guarantee can be vacuous for realistic values of $n$.

In contrast, our results offer a sharper guarantee, which covers arbitrary data distribution. Indeed, by Corollary 1, we immediately obtain

$$\log_2 \mathrm{TER}(p_{Y_0}; p_{X_0}) - \log_2 \mathrm{TER}(p_{X_0}; p_{X_0})$$

$$= \frac{1}{L}\mathsf{KL}(p_{X_0} \parallel p_{Y_0}) \leq \frac{1}{L}\mathbb{E}_M\big[\mathsf{KL}(p_{X_0} \parallel p_{Y_0|M})\big] \leq \frac{C_1}{TL}\sum_{i=1}^{L} I(X_0^{(i)}; X_0^{(-i)}) + \frac{1}{L}\varepsilon_{\mathsf{train}}. \quad (11)$$

where the first inequality makes use of the convexity of $x \mapsto -\log x$ and $p_{Y_0} = \mathbb{E}_M[p_{Y_0|M}]$. Since $I(X_0^{(i)}; X_0^{(-i)}) \leq H(X_0^{(i)}) \leq \log |\mathbb{X}|$, our KL convergence bound implies a TER bound that decays as $O((\log |\mathbb{X}|)/T)$ in the worst case. This means the token-level error in our framework drops on the order of $1/T$, regardless of $n$. Therefore, unlike Feng et al. (2025)—which is confined to specific $n$-gram distributions and degrades for high-order $n$—our bound improves the prior convergence guarantees and holds for arbitrary distributions.

## 3.2 Sampling error lower bound

Given the upper bound in Theorem 1, a natural question is whether this convergence rate can be improved. In other words, are there fundamental limits that prevent diffusion language models from converging faster than $O(1/T)$?

We proceed to answer this by establishing a matching lower bound. In fact, we prove that the dependence on the number of iterations $T$ and the mutual information in Theorem 1 is tight. In particular, Theorem 2 below provides a refined expression for the error and shows that no substantially faster rate is achievable in general. The proof can be found in Section 4.

For simplicity of presentation, we assume $\log_2 s_{\max}$ and $L/s_{\max}$ are integers without loss of generality. Otherwise, the same bounds hold up to some constant factors.

**Theorem 2.** *Consider an arbitrary mask size schedule $\{s_t\}_{t=1}^{T}$ with $s_{\max} := \max_{t \in [T]} s_t > 1$. For each token index $i \in [L]$ and integer $0 \leq j \leq \log_2 s_{\max}$, let $W_j^{(-i)} \subseteq [L]$ be a random set such that $i \notin W_j^{(-i)}$ and $|W_j^{(-i)}| = L - s_{\max}2^{-j}$. Then the output $Y_0$ of the sampling procedure* (6) *satisfies*

$$\mathbb{E}_M\big[\mathsf{KL}(p_{X_0} \parallel p_{Y_0|M})\big] \leq \frac{s_{\max}}{2L}\sum_{i=1}^{L}\sum_{j\geq 0} 2^{-j}\mathbb{E}_{W_j^{(-i)}}\big[I(X_0^{(i)}; X_0 \circ W_j^{(-i)})\big] + \varepsilon_{\mathsf{train}}. \quad (12)$$

*Moreover, there exist some mask size schedule $\{s_t\}_{t=1}^{T}$ with $s_t \asymp s_{\max}$ for all $t \in [T]$ such that*

$$\mathbb{E}_M\big[\mathsf{KL}(p_{X_0} \parallel p_{Y_0|M})\big] \geq \frac{s_{\max}}{16L}\sum_{i=1}^{L}\sum_{j\geq 0} 2^{-j}\mathbb{E}_{W_j^{(-i)}}\big[I(X_0^{(i)}; X_0 \circ W_j^{(-i)})\big] + \varepsilon_{\mathsf{train}}. \quad (13)$$

In summary, Theorem 2 demonstrates the sharpness of our analytic framework by refining the mutual information term from $\sum_{i=1}^{L} I(X_0^{(i)}; X_0^{(-i)})$ in Theorem 1 to $\sum_{i=1}^{L}\sum_{j\geq 0} 2^{-j}\mathbb{E}\big[I(X_0^{(i)}; X_0 \circ W_j^{(-i)})\big]$, which is tight up to constant factors. The somewhat complex double sum can be understood as a finer-grained decomposition of the mutual information between token $X_0^{(i)}$ and the rest of the sequence, split across different "scales" of conditioning (the sets $W_j^{(-i)}$ represent randomly chosen subsets of other tokens whose size increases as $j$ grows).

Crucially, the lower bound (13) guarantees the existence of a particular choice of $\{s_t\}_{t=1}^T$ (satisfying $s_{\max}/L \asymp 1/T$) for which the sampling error does not decay faster than on the order of $1/T$ with the same linear mutual-information dependence. In other words, it is impossible, in the worst case, to achieve a substantially smaller error than our upper bound—the $O(1/T)$ convergence rate and its linear dependence on the mutual information are fundamental limits. This matching lower bound highlights the optimality of diffusion language models' convergence analysis: we establish the best possible order of error decay for the parallel diffusion sampling scheme given the information-theoretic complexity of the text data distribution.

It is worth emphasizing that the lower bound in (13) does not hold universally for every mask size schedule. For example, if we set $s_1 = s_{\max}$ and choose $s_t = 1$ for all $t > 1$, the resulting sampling error becomes negligibly small. In this regime, a lower bound of the form (13) no longer applies. In particular, the number of iterations is $T = L + 1 - s_{\max}$, meaning the average mask size $T^{-1} \sum_{t=1}^T s_t$ is much smaller than $s_{\max}$. We conjecture that when the schedule is balanced—i.e., $T^{-1} \sum_{t=1}^T s_t \asymp s_{\max}$, as in all practical settings—matching upper and lower bounds of order $1/T$ should still be attainable. Establishing this general result is an interesting direction for future work.

**Remark 1.** Our theory provides insights into the entropy-based unmasking strategy. Specifically, (12) reveals that the per-step contribution to the total sampling error is the conditional mutual information between a newly revealed token and the remaining masked tokens. This suggests prioritizing the unmasking of tokens whose conditional dependence on the rest of the sequence is weakest. A simple heuristic to implement this strategy is to rank tokens by their conditional entropy at each step $t$: use the learned mask predictor $\widehat{p}_t(\cdot \mid Y_t)$ to estimate the conditional entropy $H(X^{(i)} \mid Y_t)$ for each masked position $i$, and unmask the positions with the lowest conditional entropy. This approach exploits the inequality $I(X; Y \mid Z) \leq H(X \mid Z)$ for any random variables $X, Y, Z$, allowing us to approximate the mutual-information criterion without requiring additional training or external estimates. Unmasking positions with lower conditional entropy thus provides a principled way to minimize the error contribution at each iteration.

## 4 Analysis

In this section, we provide the proof strategy for Theorem 1. The detailed proof for Theorem 2 and auxiliary lemmas are deferred to the appendix.

**Preparation.** We find it helpful to introduce an auxiliary sequence $(Y_t^\star)_{t=0}^T$ using the optimal mask predictor $p^\star = \prod_{i=1}^L p_i^\star$ (the minimizer of (5)). Specifically, we initialize $Y_T^\star = (\mathsf{M}, \ldots, \mathsf{M})$ and for each $t \in [T]$, define

$$Y_{t-1}^\star := \mathcal{P}_{M_t^c}(Y_t^\star) + \mathcal{P}_{M_t \setminus M_{t-1}}(X_t^\star) \quad \text{with} \quad X_t^\star \sim p^\star(\cdot \mid Y_t^\star), \qquad (14)$$

where we use the same mask schedule $(M_t)_{t=1}^T$ as in the true sampling procedure (6).

Next, let us define $W_t := M_t^c$ and $D_t := W_{t-1} \setminus W_t$ for each $t \in [T]$. By construction, $(D_t)_{t=1}^T$ forms a partition of $[L]$ and $|D_t| = s_t$ for all $t \in [T]$. Similar to $M := (M_1, \ldots, M_T)$, we denote $W := (W_1, \ldots, W_T)$ and $D := (D_1, \ldots, D_T)$ for brevity.

It is worth noting that by the construction of $(Y_t^\star)$ in (14), given a mask schedule $m = (m_1, \ldots, m_T)$, we can use the chain rule to express the distribution of $Y_0^\star$ as

$$p_{Y_0^\star \mid M}(x_0 \mid m) := p_{Y_0^\star \mid M_1, \ldots, M_T}(x_0 \mid m_1, \ldots, m_T) = \prod_{t=1}^T p^\star(x_0 \circ d_t \mid x_0 \circ w_t), \qquad (15)$$

where we recall for any set $w \in [L]$, $x \circ w$ denotes the vector in $\mathbb{X}^{|w|}$ with entries $x^{(i)}$ for $i \in w$.[5] Similarly, the distribution of the output $Y_0$ of the sampling procedure (6) given a mask schedule $m$ can be written as

$$p_{Y_0 \mid M}(x_0 \mid m) := p_{Y_0 \mid M_1, \ldots, M_T}(x_0 \mid m_1, \ldots, m_T) = \prod_{t=1}^T \widehat{p}(x_0 \circ d_t \mid x_0 \circ w_t). \qquad (16)$$

---

[5]Here and throughout this paper, we slightly abuse the notation: in (15), we write $p^\star(x_0 \circ d_t \mid x_0 \circ w_t)$ in a way that it accepts an input of length $|w_t|$, while $p^\star$, defined in (5), takes a masked sequence of length $L$. It is not hard to see that the two are equivalent since the remaining tokens are replaced by the mask $\mathsf{M}$.

With the above preparation complete, we now begin to prove Theorem 1.

**Step 1: Decoupling training error.** We begin by separating the training error from the fundamental sampling difficulty. For any mask schedule $m$, we can write:

$$\mathsf{KL}\big(p_{X_0}(\cdot) \,\|\, p_{Y_0|M}(\cdot \mid m)\big) - \mathsf{KL}\big(p_{X_0}(\cdot) \,\|\, p_{Y_0^\star|M}(\cdot \mid m)\big)$$

$$= \int_{\mathbb{X}^L} p_{X_0}(x_0) \log \frac{p_{Y_0^\star|M}(x_0 \mid m)}{p_{Y_0|M}(x_0 \mid m)} \,\mathrm{d}x_0 \overset{\text{(i)}}{=} \sum_{t=1}^{T} \int_{\mathbb{X}^L} p_{X_0}(x_0) \log \frac{p^\star(x_0 \circ d_t \mid x_0 \circ w_t)}{\widehat{p}(x_0 \circ d_t \mid x_0 \circ w_t)} \,\mathrm{d}x_0$$

$$\overset{\text{(ii)}}{=} \sum_{t=1}^{T} \int_{\mathbb{X}^L} p_{X_0}(x_0) \sum_{i \in d_t} \log \frac{p^\star(x_0^{(i)} \mid x_0 \circ w_t)}{\widehat{p}(x_0^{(i)} \mid x_0 \circ w_t)} \,\mathrm{d}x_0$$

$$\overset{\text{(iii)}}{=} \mathbb{E}_{\tau, X_0} \left[ \frac{L}{s_\tau} \sum_{i \in D_\tau} \log \frac{p_i^\star(X_0^{(i)} \mid X_0 \circ W_\tau)}{\widehat{p}_i(X_0^{(i)} \mid X_0 \circ W_\tau)} \,\middle|\, M = m \right],$$

Here, (i) follows from $p_{Y_0|M}(x_0 \mid m) = \prod_{t=1}^{T} \widehat{p}(x_0 \circ d_t \mid x_0 \circ w_t)$ and $p_{Y_0^\star|M}(x_0 \mid m) = \prod_{t=1}^{T} p^\star(x_0 \circ d_t \mid x_0 \circ w_t)$ as shown in (16) and (15), respectively; (ii) is true as $p^\star$ and $\widehat{p}$ are product distributions; (iii) holds because $\mathbb{P}\{\tau = t\} = s_t/L$. Since each set $D_t$ of size $s_t$ represents the positions newly unmasked at step $t$, which are chosen uniformly at random from the previously masked positions $M_t = W_t^c$, taking expectations over all mask realizations yields:

$$\mathbb{E}_M\big[\mathsf{KL}(p_{X_0} \,\|\, p_{Y_0|M}) - \mathsf{KL}(p_{X_0} \,\|\, p_{Y_0^\star|M})\big] = \mathbb{E}_{\tau, X_0, M_\tau} \left[ \frac{L}{|M_\tau|} \sum_{i \in M_\tau} \log \frac{p^\star(X_0^{(i)} \mid X_0 \circ W_\tau)}{\widehat{p}(X_0^{(i)} \mid X_0 \circ W_\tau)} \right] = \varepsilon_{\text{train}}. \tag{17}$$

where the last step follows from the definition of $\varepsilon_{\text{train}}$ in (7).

This decomposition shows that in order to control the KL divergence $\mathbb{E}_M[\mathsf{KL}(p_{X_0} \,\|\, p_{Y_0|M})]$ between the distributions of the output $Y_0$ and data $X_0$, it suffices to focus on the KL divergence $\mathbb{E}_M[\mathsf{KL}(p_{X_0} \,\|\, p_{Y_0^\star|M})]$ between the distributions of the auxiliary output $Y_0^\star$ and data $X_0$.

**Step 2: Parameterizing by maximum mask size.** Our strategy is to establish a recursive inequality that relates the sampling performance with maximum mask size $s_{\max}$ to those with smaller mask sizes. Towards this, recall that the sizes of the mask sets $\{M_t\}_{t=1}^T$ are determined by the mask size schedule $\{s_t\}_{t=1}^T$. To establish our recursive bound, we parameterize the sampling difficulty by the maximum mask size. Concretely, we define

$$\varepsilon(s_{\max}) := \max_{\{s_t\}_{t=1}^T:\, \max_{t \in [T]} s_t = s_{\max}} \varepsilon(\{s_t\}), \tag{18a}$$

where for any mask size schedule $\{s_t\}_{t=1}^T$, define

$$\varepsilon(\{s_t\}) := \mathbb{E}_M\big[\mathsf{KL}(p_{X_0} \,\|\, p_{Y_0^\star|M})\big], \tag{18b}$$

We now present the following key lemma that controls $\varepsilon(s_{\max})$; with proof deferred to Appendix A.

**Lemma 1.** *For any $s_{\max} > 1$, one has*

$$\varepsilon(s_{\max}) \leq \varepsilon(\lceil s_{\max}/2 \rceil) + \frac{s_{\max}}{2L} \sum_{i=1}^{L} I(X_0^{(i)}; X_0^{(-i)}). \tag{19}$$

Given Lemma 1, we can apply the inequality (19) recursively to obtain

$$\varepsilon(s_{\max}) \leq \varepsilon(1) + \sum_{j=0}^{\lceil \log_2 s_{\max} \rceil - 1} \frac{2^j}{L} \sum_{i=1}^{L} I(X_0^{(i)}; X_0^{(-i)}) = \varepsilon(1) + \frac{2^{\lceil \log_2 s_{\max} \rceil} - 1}{L} \sum_{i=1}^{L} I(X_0^{(i)}; X_0^{(-i)}). \tag{20}$$

Finally, note that when the maximum mask size is equal to 1, we have $|M_t \setminus M_{t-1}| = 1$ for all $t \in [T]$, i.e., the sampling process unmasks tokens one by one. In this case, it is not hard to use the construction of $Y_0^\star$ in (14) and the fact that $p_i^\star(\cdot \mid X_t) = p_{X_0^{(i)}|X_t}(\cdot \mid X_t)$ to deduce that $\varepsilon(1) = 0$. The claim (8) then immediately follows from (17) and (20).

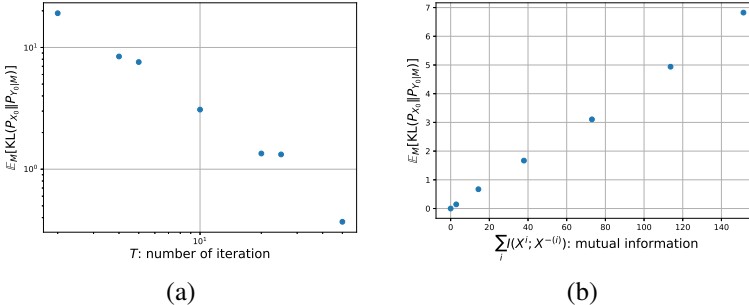

(a)                                                        (b)

Figure 1: (a) sampling error vs. number of iterations $T$ where $J = 2$; (b) sampling error vs. mutual information where $T = 10$.

## 5  Numerical Experiments

In this section, we present numerical experiments to validate our convergence theory developed in Section 3. For the data distribution $p_{\text{data}}$ of text $X = (X^{(1)}, \ldots, X^{(L)})$, we consider a $K$-state Potts chain of length $L$ with coupling parameter $J$. Specifically, $X^{(1)} \sim \text{Unif}([K])$ and for $i \geq 2$,

$$\mathbb{P}\{X^{(i)} = y \mid X^{(i-1)} = x\} = \frac{\exp\big(J\,\mathbb{1}\{x = y\}\big)}{\exp(J) + K - 1}, \qquad \forall\, x, y \in [K].$$

This construction allows us to compute explicitly the mutual information $I(X^{(i)}; X^{(-i)})$, the optimal mask predictor $p^*(\cdot \mid X_t)$, and the distributions of both the data $p_{X_0}$ and the generated sample $p_{Y_0 \mid M}$. We implement the sampling process using the optimal mask predictor $p^*(\cdot \mid X_t)$ and a balanced mask schedule where the number of unmasked tokens is the same at each iteration. Given the explicit distributions of $p_{X_0}$ and $p_{Y_0 \mid M}$, the expectation in the KL divergence, taken over both the mask schedule $M$ and the data distribution $p_{X_0}$, is approximated via Monte Carlo simulations.

Set $K = 10$ and $L = 100$. Figure 1 (a) presents the sampling error (in KL divergence) vs. the number of iterations $T$. As shown, the slope in the log-log plot is very close to $-1$, demonstrating that the sampling error scales proportionally to $1/T$. In addition, Figure 1 (b) plots the KL sampling error vs. the mutual information (controlled by $J$). One can see that the sampling error increases approximately linearly with the mutual information. Collectively, these numerical studies confirm our main theoretical findings: the KL sampling error decays as $O(1/T)$ with the number of iterations and grows linearly with mutual information.

## 6  Discussion

In this work, we have made progress towards understanding the sampling process in diffusion language models. Our results provide tight convergence guarantees, revealing that the sampling error—quantified by the KL divergence—decreases on the order of $1/T$ with the number of iterations and increases linearly with the mutual information among tokens.

Looking ahead, our analysis suggests that the sampling error primarily stems from the discrepancy between the true data distribution and the modeled product distribution. This observation motivates future studies to explore low-dimensional structures or low-order Markov properties in the text data, which may help reduce this discrepancy and thereby decrease the sampling error. Moreover, establishing comprehensive end-to-end performance guarantees that account for both the mask training phase and the sampling phase represents an important direction for further research. Finally, while our current focus is on masked diffusion models, extending these insights to other types of discrete diffusion models for language modeling is a compelling avenue for future investigation.

## Acknowledgments and Disclosure of Funding

G. Li is supported in part by the Chinese University of Hong Kong Direct Grant for Research and the Hong Kong Research Grants Council ECS 2191363. C. Cai is supported in part by the NSF grant DMS-2515333.

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

# A    Proof of Lemma 1

Before proving the inequality (19), we first introduce some notation and make a few preliminary observations.

Recall that $X \circ W \in |\mathbb{X}|^{|W|}$ denote the sub-vector of $X$ indexed by set $W$. For simplicity of presentation, for any set $W \subseteq [L]$, we denote by

$$p(\cdot \mid X_0 \circ W) := p_{X_0|X_0 \circ W}(\cdot \mid X_0 \circ W)$$

the conditional distribution of $X_0$ given the partial tokens $X_0 \circ W$. Moreover, we define the associated product distribution over $\mathbb{X}^L$ as

$$p^{\otimes}(\cdot \mid X_0 \circ W) := \prod_{i=1}^{L} p_i(\cdot \mid X_0 \circ W) \quad \text{with} \quad p_i(\cdot \mid X_0 \circ W) := p_{X_0^{(i)}|X_0 \circ W}(\cdot \mid X_0 \circ W), \ i \in [L].$$

In a word, $p_i(\cdot \mid X_0 \circ W)$ denotes the conditional distribution of the $i$-th coordinate given the partial tokens $X_0 \circ W$ and the product distribution $p^{\otimes}(\cdot \mid X_0 \circ W)$ treats all coordinates as conditionally independent.

Next, recall that for any mask schedule $M = (M_1, \ldots, M_T)$, we denote $W_t := M_t^c$ and $D_t := W_{t-1} \setminus W_t$. Since $\{D_t\}_{t=1}^T$ forms a partition of $[L]$, we know from the chain rule that given a mask schedule $M$, the distribution of $X_0$ can be factorized as

$$p_{X_0|M}(X_0 \mid M) = \prod_{t=1}^{T} p(X_0 \circ D_t \mid X_0 \circ W_t). \tag{21}$$

Meanwhile, recall that the "ideal" sampling procedure (14) yields that $p_{Y_0^\star|M}(x_0 \mid m) = \prod_{t=1}^{T} p^\star(x_0 \circ d_t \mid x_0 \circ w_t)$ and that the minimizer $p_i^\star$ of (5) coincides with $p_i(\cdot \mid X \circ W)$. Thus, given the mask sequence $M$, the distribution of the auxiliary output $Y_0^\star$ of (14) can be expressed as

$$p_{Y_0^\star|M}(X_0 \mid M) = \prod_{t=1}^{T} p^{\otimes}(X_0 \circ D_t \mid X_0 \circ W_t). \tag{22}$$

Putting the above observations together implies given a mask sequence $M$, the KL divergence between the data $X_0$ and the auxiliary output $Y_0^\star$ can be decomposed as a sum of KL divergences between the true conditionals and their product-form counterparts at each sampling step:

$$\mathbb{E}_M\big[\mathsf{KL}(p_{X_0} \parallel p_{Y_0^\star|M})\big] = \sum_{t=1}^{T} \mathbb{E}_M\Big[\mathsf{KL}\big(p(X_0 \circ D_t \mid X_0 \circ W_t) \parallel p^{\otimes}(X_0 \circ D_t \mid X_0 \circ W_t)\big)\Big]. \tag{23}$$

With the above preparation in place, we are now ready to prove the inequality (19). In light of the definitions of $\varepsilon(s_{\max})$ and $\varepsilon(\{s_t\})$ in (18), it suffices to study the KL divergence term on the right-hand side of (23) for each $t \in [T]$.

To this end, let us fix an arbitrary mask size schedule $\{s_t\}_{t=1}^T$ with $\max_{t \in [T]} s_t = s_{\max}$ and consider a mask set sequence $M = (M_1, \ldots, M_T)$ such that $|M_t \setminus M_{t-1}| = s_t$. We construct an intermediate mask schedule whose maximum mask size equals $\lceil s_{\max}/2 \rceil$. Specifically, for each $t \in [T]$, let $W_{t-1/2}$ be a random set such that $W_t \subseteq W_{t-1/2} \subseteq W_{t-1}$ and $W_{t-1/2} \setminus W_t$ is a random subset of $D_t = W_{t-1} \setminus W_t$ with size $\lceil s_t/2 \rceil$. For notional convenience, we define the following two sets:

$$D_{t,-} := W_{t-1/2} \setminus W_t \qquad \text{(first set, size } \lceil s_t/2 \rceil)$$
$$D_{t,+} := W_{t-1} \setminus W_{t-1/2} \qquad \text{(second set, size } \lfloor s_t/2 \rfloor)$$

The key insight is that revealing $D_t = D_{t,-} \cup D_{t,+}$ in two stages creates a dependency structure that we can exploit. Conditioned on $M = m$, we can express each KL divergence on the right-hand-side

of (23) as follows:

$$
\begin{aligned}
&\mathsf{KL}\big(p(X_0 \circ d_t \mid X_0 \circ w_t) \,\|\, p^{\otimes}(X_0 \circ d_t \mid X_0 \circ w_t)\big)\\
&\overset{\text{(i)}}{=} \mathsf{KL}\big(p(X_0 \circ d_{t,-} \mid X_0 \circ w_t)p(X_0 \circ d_{t,+} \mid X_0 \circ w_{t-1/2})\\
&\qquad\quad \|\, p^{\otimes}(X_0 \circ d_{t,-} \mid X_0 \circ w_t)p^{\otimes}(X_0 \circ d_{t,+} \mid X_0 \circ w_t)\big)\\
&\overset{\text{(ii)}}{=} \mathsf{KL}\big(p(X_0 \circ d_{t,-} \mid X_0 \circ w_t) \,\|\, p^{\otimes}(X_0 \circ d_{t,-} \mid X_0 \circ w_t)\big)\\
&\quad + \mathbb{E}_{X_0 \circ d_{t,-}}\big[\mathsf{KL}\big(p(X_0 \circ d_{t,+} \mid X_0 \circ w_{t-1/2}) \,\|\, p^{\otimes}(X_0 \circ d_{t,+} \mid X_0 \circ w_t)\big) \mid X_0 \circ w_t\big]\\
&\overset{\text{(iii)}}{=} \mathsf{KL}\big(p(X_0 \circ d_{t,-} \mid X_0 \circ w_t) \,\|\, p^{\otimes}(X_0 \circ d_{t,-} \mid X_0 \circ w_t)\big)\\
&\quad + \mathbb{E}_{X_0 \circ d_{t,-}}\big[\mathsf{KL}\big(p(X_0 \circ d_{t,+} \mid X_0 \circ w_{t-1/2}) \,\|\, p^{\otimes}(X_0 \circ d_{t,+} \mid X_0 \circ w_{t-1/2})\big) \mid X_0 \circ w_t\big]\\
&\quad + \sum_{i \in d_{t,+}} I(X_0^{(i)}; X_0 \circ d_{t,-} \mid X_0 \circ w_t). \qquad\qquad\qquad\qquad\qquad\qquad\qquad (24)
\end{aligned}
$$

Here, (i) holds as $D_t \setminus D_{t,-} = D_{t,+}$ and $W_{t-1/2} \setminus W_t = D_{t,-}$; (ii) applies the chain rule of the KL divergence; (iii) makes use of the following identity:

$$
\begin{aligned}
&\int p(X_0 \circ d_{t,-} \mid X_0 \circ w_t)\, p(X_0 \circ d_{t,+} \mid X_0 \circ w_{t-1/2}) \log \frac{p^{\otimes}(X_0 \circ d_{t,+} \mid X_0 \circ w_{t-1/2})}{p^{\otimes}(X_0 \circ d_{t,+} \mid X_0 \circ w_t)}\\
&\overset{\text{(i)}}{=} \sum_{i \in d_{t,+}} \int p(X_0 \circ d_{t,-} \mid X_0 \circ w_t)\, p(X_0 \circ d_{t,+} \mid X_0 \circ w_{t-1/2}) \log \frac{p^{\otimes}(X_0^{(i)} \mid X_0 \circ w_{t-1/2})}{p^{\otimes}(X_0^{(i)} \mid X_0 \circ w_t)}\\
&\overset{\text{(ii)}}{=} \sum_{i \in d_{t,+}} \int p(X_0 \circ d_{t,-} \mid X_0 \circ w_t)\, p^{\otimes}(X_0^{(i)} \mid X_0 \circ w_{t-1/2}) \log \frac{p^{\otimes}(X_0^{(i)} \mid X_0 \circ w_{t-1/2})}{p^{\otimes}(X_0^{(i)} \mid X_0 \circ w_t)}\\
&\overset{\text{(iii)}}{=} \sum_{i \in d_{t,+}} \int p(X_0 \circ d_{t,-} \mid X_0 \circ w_t)\, p^{\otimes}(X_0^{(i)} \mid X_0 \circ d_{t,-}, X_0 \circ w_t) \log \frac{p^{\otimes}(X_0^{(i)} \mid X_0 \circ d_{t,-}, X_0 \circ w_t)}{p^{\otimes}(X_0^{(i)} \mid X_0 \circ w_t)}\\
&= \sum_{i \in d_{t,+}} I(X_0^{(i)}; X_0 \circ d_{t,-} \mid X_0 \circ w_t).
\end{aligned}
$$

where (i) follows from our construction of the product distribution $p^{\otimes}$; (ii) is true as the marginal distributions of $p(X_0 \circ d_{t,+} \mid X_0 \circ w_{t-1/2})$ and $p^{\otimes}(X_0 \circ d_{t,+} \mid X_0 \circ w_{t-1/2})$ are identical; (iii) holds because $W_t \cap D_{t,-} = \varnothing$ and $W_t \cup D_{t,-} = W_{t-1/2}$.

Notice that in (24), the last term captures the dependency between the two sets of tokens while the first two terms correspond to a sampling process with maximum mask size $\lceil s_{\max}/2 \rceil$.

Putting (23) and (24) together with the definition of $\varepsilon(s_{\max})$ in (18), we can derive

$$
\varepsilon(s_{\max}) \le \varepsilon(\lceil s_{\max}/2 \rceil) + \mathbb{E}_W\left[\sum_{t=1}^{T} \sum_{i \in D_{t,+}} I(X_0^{(i)}; X_0 \circ D_{t,-} \mid X_0 \circ W_t)\right]. \qquad (25)
$$

For the mutual information term, taking the expectation with respect to $W = (W_1, \ldots, W_T)$ (or equivalently $M = (M_1, \ldots, M_T)$) and summing over $t = 1, \ldots, T$ yields

$$\mathbb{E}_W \left[ \sum_{t=1}^{T} \sum_{i \in D_{t,+}} I(X_0^{(i)}; X_0 \circ D_{t,-} \mid X_0 \circ W_t) \right]$$

$$\stackrel{(i)}{=} \sum_{t=1}^{T} \left\lfloor \frac{s_t}{2} \right\rfloor \mathbb{E}_{W_t, W_{t-1/2}, i \sim \mathsf{Unif}(W_{t-1/2}^c)} \left[ I(X_0^{(i)}; X_0 \circ D_{t,-} \mid X_0 \circ W_t) \right]$$

$$= \sum_{t=1}^{T} \frac{1}{L} \left\lfloor \frac{s_t}{2} \right\rfloor \sum_{i=1}^{L} \mathbb{E}_{W_t, W_{t-1/2}} \left[ I(X_0^{(i)}; X_0 \circ D_{t,-} \mid X_0 \circ W_t) \mid i \notin W_{t-1/2} \right]$$

$$\leq \frac{s_{\max}}{2L} \sum_{i=1}^{L} \mathbb{E}_W \left[ \sum_{t=1}^{T} I(X_0^{(i)}; X_0 \circ D_{t,-} \mid X_0 \circ W_t) \mid i \notin W_{1/2} \right]$$

$$\stackrel{(ii)}{\leq} \frac{s_{\max}}{2L} \sum_{i=1}^{L} I(X_0^{(i)}; X_0^{(-i)}), \tag{26}$$

where (i) is true because $D_{t,+}$ is a random subset of $W_{t-1/2}^c$ with $|D_{t,+}| = \lfloor s_t/2 \rfloor$; (ii) arises from the following bound:

$$\mathbb{E}_W \left[ \sum_{t=1}^{T} I(X_0^{(i)}; X_0 \circ D_{t,-} \mid X_0 \circ W_t) \mid i \notin W_{1/2} \right] \leq I(X_0^{(i)}; X_0^{(-i)})$$

due to $W_{t-1} \setminus W_t = D_t = D_{t,-} \cup D_{t,+}$ and the chain rule of mutual information that $I(X; Y \mid Z) + I(X; Z) = I(X; Y, Z)$ for any $X, Y, Z \sim p_{X,Y,Z}$.

Combining (25) and (26) establishes the recursive inequality (19), thereby completing the proof of Theorem 1.

## B   Proof of Theorem 2

In this section, we prove Theorem 2. Our strategy is to establish the lower bound (13) first, then sharpen the factor in the upper bound (8) to obtain the refined upper bound (12).

### B.1   Lower bound analysis

We begin by reminding the readers of the sampling process introduced in Section 2. Recall that $M_t$ denotes the set of masked positions at step $t$ and that we define $W_t := [L] \setminus M_t$ as the set of unmasked positions. Equivalently, the sampling process creates a decreasing sequence of random sets $[L] = W_0 \supseteq W_1 \supseteq \cdots \supseteq W_T = \varnothing$, where each $W_t$ is obtained from $W_{t-1}$ by removing $s_t$ newly revealed positions. The sampler starts with a fully masked sequence $Y_T = (\mathsf{M}, \ldots, \mathsf{M})$ and iteratively reveals tokens by going backwards through time $t = T, T-1, \ldots, 1$. At each step $t$, the sampler predicts $s_t$ tokens located in the unmask set $W_{t-1} \setminus W_t$.

**Step 1: Auxiliary sampling process.**   To establish the lower bound, let us consider a specific mask size schedule $\{s_t\}_{t=1}^{T}$. For some $s_{\max} > 1$, each $s_t$ is independently chosen from $\{s_{\max}, s_{\max}/2\}$ uniformly at random. Without loss of generality, we assume that $L = \sum_{t=1}^{T} s_t$, which implies that $T = (1 + o(1)) \frac{2L}{3 s_{\max}}$.

To analyze the sampling process with the chosen mask size schedule, we reorganize the original $T$-step sampling process into a $K$-step process where $K := 2L/s_{\max}$. Let $[L] = W_0 \supseteq W_1 \supseteq \cdots \supseteq W_K = \varnothing$ be a decreasing unmask sets where each $W_k$ is a random subset of $W_{k-1}$ such that $|W_{k-1} \setminus W_k| = s_{\max}/2$. In this reorganized view, each "super-step" in the $K$-step process corresponds to revealing $s_{\max}/2$ positions. The correspondence between original steps and super-steps is as follows:

- When $s_t = s_{\max}/2$ in the original process: the auxiliary sampler takes one super-step $(k \to k-1)$.

- When $s_t = s_{\max}$ in the original process: the auxiliary sampler takes two super-steps at once $(k \to k-2)$.

Since each $s_t$ is chosen uniformly from $\{s_{\max}, s_{\max}/2\}$, each type of transition occurs with probability $1/2$.

The key insight comes from analyzing two-super-step transitions $(k \to k-2)$, which occur when $s_t = s_{\max}$. Consider the case where the sampling process transitions from $k$ to $k-2$, which happens with probability at least $1/4$. For such transitions, define:

$$
\begin{aligned}
D_k &:= W_{k-2} \setminus W_k, && \text{(all newly revealed positions)} \\
D_{k,-} &:= W_{k-1} \setminus W_k, && \text{(first batch, size } s_{\max}/2) \\
D_{k,+} &:= W_{k-2} \setminus W_{k-1}. && \text{(second batch, size } s_{\max}/2)
\end{aligned}
$$

Using the non-negativity of the KL divergence and repeating the argument for (26), we obtain the following lower bound:

$$
\mathbb{E}_M\big[\mathsf{KL}(p_{X_0} \| p_{Y_0|M})\big] - \varepsilon_{\mathsf{train}} = \mathbb{E}_M\big[\mathsf{KL}(p_{X_0} \| p_{Y_0^\star|M})\big]
$$

$$
\geq \frac{1}{4} \sum_{k=1}^K \mathbb{E}\bigg[\sum_{i \in D_{k,+}} I(X_0^{(i)}; X_0 \circ D_{k,-} \mid X_0 \circ W_k)\bigg]
$$

$$
= \frac{s_{\max}}{8L} \sum_{i=1}^L \sum_{k=1}^K \mathbb{E}\Big[I(X_0^{(i)}; X_0 \circ D_{k,-} \mid X_0 \circ W_k) \mid i \notin W_1\Big]
$$

$$
= \frac{s_{\max}}{8L} \sum_{i=1}^L \mathbb{E}\big[I(X_0^{(i)}; X_0 \circ W_1) \mid i \notin W_1\big]. \tag{27}
$$

**Step 2: Hierarchical decomposition.** In what follows, we will develop a stronger lower bound through a more sophisticated recursive analysis, which leads to the desired result (13). To this end, for any super-step $k$ with two-step transition, applying the decomposition in (24) and the non-negativity of the KL divergence, we can derive: conditioned on $W = w$,

$$
\mathsf{KL}\big(p(X_0 \circ d_k \mid X_0 \circ w_k) \| p^\otimes(X_0 \circ d_k \mid X_0 \circ w_k)\big)
$$

$$
\geq \sum_{i \in d_{k,+}} I(X_0^{(i)}; X_0 \circ d_{k,-} \mid X_0 \circ w_k)
$$

$$
+ \mathbb{E}_{X_0 \circ d_{k,-}}\big[\mathsf{KL}\big(p(X_0 \circ d_{k,+} \mid X_0 \circ w_{k-1}) \| p^\otimes(X_0 \circ d_{k,+} \mid X_0 \circ w_{k-1})\big)\big]. \tag{28}
$$

Consider the case $k = 2$ where the sampler uses $W_2$ and $W_0$ consecutively. The above inequality (28) tells us that

$$
\mathbb{E}_W\big[\mathsf{KL}\big(p(X_0 \circ D_2 \mid X_0 \circ W_2) \| p^\otimes(X_0 \circ D_2 \mid X_0 \circ W_2)\big)\big]
$$

$$
\geq \mathbb{E}_W\bigg[\sum_{i \in D_{2,+}} I(X_0^{(i)}; X_0 \circ D_{2,-} \mid X_0 \circ W_2)\bigg]
$$

$$
+ \mathbb{E}_{W, X_0 \circ D_{2,-}}\big[\mathsf{KL}\big(p(X_0 \circ D_{2,+} \mid X_0 \circ W_1) \| p^\otimes(X_0 \circ D_{2,+} \mid X_0 \circ W_1)\big)\big].
$$

By construction, one has $|W_2| = L - s_{\max}$, $|W_1| = L - s_{\max}/2$, and $|D_{2,-}| = |D_{2,+}| = s_{\max}/2$.

To leverage this structure, we define a hierarchical family of random sets: for any $i \in [L]$, let $\widehat{W}_0^{(-i)} \subseteq \cdots \subseteq \widehat{W}_j^{(-i)} \subseteq \cdots \subseteq [L]$ be a sequence of increasing random sets such that $i \notin \widehat{W}_j^{(-i)}$ and $|\widehat{W}_j^{(-i)}| = L - s_{\max} 2^{-j}$ for all $0 \leq j \leq \log_2 s_{\max}$. Consequently, we find that

$$
\mathbb{E}_W\big[\mathsf{KL}\big(p(X_0 \circ D_2 \mid X_0 \circ W_2) \| p^\otimes(X_0 \circ D_2 \mid X_0 \circ W_2)\big)\big]
$$

$$
\stackrel{(i)}{\geq} \frac{s_{\max}}{2} \mathbb{E}_{\widehat{W}_1^{(-i)}, \widehat{W}_0^{(-i)}}\big[I(X_0^{(i)}; X_0 \circ \widehat{W}_1^{(-i)} \mid X_0 \circ \widehat{W}_0^{(-i)})\big]
$$

$$
+ \mathbb{E}_{W, X_0 \circ D_{2,-}}\big[\mathsf{KL}\big(p(X_0 \circ D_{2,+} \mid X_0 \circ W_1) \| p^\otimes(X_0 \circ D_{2,+} \mid X_0 \circ W_1)\big)\big]
$$

where the inequality holds as $\widehat{W}_0^{(-i)} \subseteq \widehat{W}_1^{(-i)}$ and $|\widehat{W}_1^{(-i)} \setminus \widehat{W}_0^{(-i)}| = |D_{2,-}| = |D_{2,+}| = s_{\max}/2$. Applying the above relationship recursively across all hierarchical levels and invoking the decomposition (23) yields

$$\mathbb{E}_M\big[\mathsf{KL}(p_{X_0} \parallel p_{Y_0^\star|M})\big] \gtrsim \frac{s_{\max}}{L} \sum_{i=1}^{L} \sum_{j=1}^{\log_2 s_{\max}} 2^{-j} \mathbb{E}_{\widehat{W}_j^{(-i)}, \widehat{W}_{j-1}^{(-i)}} \big[I(X_0^{(i)}; X_0 \circ \widehat{W}_j^{(-i)} \mid X_0 \circ \widehat{W}_{j-1}^{(-i)})\big].$$
(29)

Now we simplify the hierarchical sum on the right-hand side of (29). Recall that for any $i \in [L]$ and $j \geq 0$, we define $W_j^{(-i)} \subseteq [L]$ to be a random set such that $i \notin W_j^{(-i)}$ and $|W_j^{(-i)}| = L - s_{\max} 2^{-j}$. Combining $\{W_j^{(-i)}\}_{j \geq 1}$ with $\{\widehat{W}_j^{(-i)}\}_{j \geq 1}$, we can derive

$$\sum_{j=1}^{\log_2 s_{\max}} 2^{-j} \mathbb{E}_{\widehat{W}_j^{(-i)}, \widehat{W}_{j-1}^{(-i)}} \big[I(X_0^{(i)}; X_0 \circ \widehat{W}_j^{(-i)} \mid X_0 \circ \widehat{W}_{j-1}^{(-i)})\big]$$

$$\overset{(i)}{=} \sum_{j=1}^{\log_2 s_{\max}} 2^{-j} \mathbb{E}_{\widehat{W}_j^{(-i)}, \widehat{W}_{j-1}^{(-i)}} \big[I(X_0^{(i)}; X_0 \circ \widehat{W}_j^{(-i)}) - I(X_0^{(i)}; X_0 \circ \widehat{W}_{j-1}^{(-i)})\big]$$

$$\overset{(ii)}{=} \sum_{j=1}^{\log_2 s_{\max}} 2^{-j} \mathbb{E}_{W_j^{(-i)}} \big[I(X_0^{(i)}; X_0 \circ W_j^{(-i)})\big] - \frac{1}{2} \sum_{j=1}^{\log_2 s_{\max}} 2^{-(j-1)} \mathbb{E}_{W_{j-1}^{(-i)}} \big[I(X_0^{(i)}; X_0 \circ W_{j-1}^{(-i)})\big]$$

$$= \frac{1}{2} \sum_{j=1}^{\log_2 s_{\max}} 2^{-j} \mathbb{E}_{W_j^{(-i)}} \big[I(X_0^{(i)}; X_0 \circ W_j^{(-i)})\big] - \frac{1}{2} \mathbb{E}_{W_0^{(-i)}} \big[I(X_0^{(i)}; X_0 \circ W_0^{(-i)})\big]$$

$$+ \frac{1}{s_{\max}} \mathbb{E}_{W_{\log_2 s_{\max}}^{(-i)}} \big[I(X_0^{(i)}; X_0 \circ W_{\log_2 s_{\max}}^{(-i)})\big]$$

$$\geq \frac{1}{2} \sum_{j=1}^{\log_2 s_{\max}} 2^{-j} \mathbb{E}_{W_j^{(-i)}} \big[I(X_0^{(i)}; X_0 \circ W_j^{(-i)})\big] - \frac{1}{2} \mathbb{E}_{W_0^{(-i)}} \big[I(X_0^{(i)}; X_0 \circ W_0^{(-i)})\big].$$

where (i) uses the chain rule of the mutual information; (ii) holds as $W_j^{(-i)}$ and $\widehat{W}_j^{(-i)}$ have the same marginal distribution. Substituting the above bound into (29), we obtain

$$\mathbb{E}_M\big[\mathsf{KL}(p_{X_0} \parallel p_{Y_0^\star|M})\big] \gtrsim \frac{s_{\max}}{L} \sum_{i=1}^{L} \bigg\{ \sum_{j=1}^{\log_2 s_{\max}} 2^{-j} \mathbb{E}_{W_j^{(-i)}} \big[I(X_0^{(i)}; X_0 \circ W_j^{(-i)})\big] - \mathbb{E}_{W_0^{(-i)}} \big[I(X_0^{(i)}; X_0 \circ W_0^{(-i)})\big] \bigg\}.$$
(30)

**Step 3: Combining bounds.** Finally, it is not hard to deduce from the basic bound (27) that

$$\mathbb{E}_M\big[\mathsf{KL}(p_{X_0} \parallel p_{Y_0^\star|M})\big] \gtrsim \frac{s_{\max}}{L} \sum_{i=1}^{L} \mathbb{E}_{W_0^{(-i)}} \big[I(X_0^{(i)}; X_0 \circ W_0^{(-i)})\big].$$
(31)

Therefore, combining (30) and (31) with the training error bound (17) yields the desired lower bound (13).

## B.2 Upper bound analysis

For the refined upper bound (12), we will use the introduced random sets $\{W_j^{(-i)}\}_{j \geq 1}$ to improve the analysis in step (ii) of (26). Since $W_{t-1} \setminus W_t = D_t = D_{t,-} \cup D_{t,+}$, one can use the chain rule of the mutual information to derive

$$\mathbb{E}_W\bigg[ \sum_{t=1}^{T} \sum_{i \in D_{t,+}} I(X_0^{(i)}; X_0 \circ D_{t,-} \mid X_0 \circ W_t) \bigg] \leq \frac{s_{\max}}{2L} \sum_{i=1}^{L} \mathbb{E}_{W_0^{(-i)}} \big[I(X_0^{(i)}; X_0 \circ W_0^{(-i)})\big],$$
(32)

where we recall $W_0^{(-i)} \subseteq [L]$ to be a random set such that $i \notin W_0^{(-i)}$ and $|W_0^{(-i)}| = L - s_{\max}$. Hence, applying the same recursive argument as for (29), this improvement allows us to obtain the refined inductive relationship (19) as follows. For any $0 \leq j < \log_2 s_{\max}$:

$$\varepsilon(s_{\max}2^{-j}) \leq \varepsilon(s_{\max}2^{-(j+1)}) + \frac{s_{\max}}{2L}2^{-j}\sum_{i=1}^{L}\mathbb{E}\big[I(X_0^{(i)}; X_0 \circ W_j^{(-i)})\big]. \tag{33}$$

Applying this inequality recursively gives

$$\varepsilon(s_{\max}) \leq \varepsilon(1) + \frac{s_{\max}}{2L}\sum_{j=0}^{\log_2 s_{\max}-1}2^{-j}\sum_{i=1}^{L}\mathbb{E}\big[I(X_0^{(i)}; X_0 \circ W_j^{(-i)})\big]. \tag{34}$$

Therefore, the desired refined upper bound (12) immediately follows from the fact that $\varepsilon(1) = 0$.

