# OpenReview forum: "Breaking AR’s Sampling Bottleneck: Provable Acceleration via Diffusion Language Models"
_NeurIPS.cc/2025/Conference — NeurIPS 2025 poster_

### Official Review · Reviewer_v9hs · 2025-06-25

**Clarity:** 3
**Significance:** 2
**Originality:** 3
**Rating:** 5
**Confidence:** 3

**Summary:**

This manuscript proves convergence guarantees for sampling from discrete diffusion language models.  The authors prove that (i) the error decays as O(1/T) after T denoising steps and (ii) that the constant factor depends on the mutual information between tokens in the target sequence. Finally, a matching lower bound shows that this rate is tight. The results assume access to accurate mask predictors, consistent with prior work that theoretically analyze the sampling process of continuous diffusion models.

**Questions:**

1. Can the bound be tightened further for language distributions with known structure (e.g., low-order Markov texts)? Have the authors thought about interesting such distributions?
2. Have the authors studied the impact of changing the sampling algorithm to the bound? For example, what would be the impact of entropy-based sampling, where positions with low uncertainty (as predicted by the denoiser), are more likely to be sampled first?

**Ethical Concerns:**

["NO or VERY MINOR ethics concerns only"]

**Final Justification:**

The authors have addressed my questions and even included small-scale experiments, despite being primarily a theoretical contribution. I think this paper makes a valuable contribution to the literature, by analyzing a realistic model of discrete diffusion models. Hence, I think the paper should be accepted

**Limitations:**

yes

**Quality:**

3

**Strengths And Weaknesses:**

## Strengths
- **Well-written**: in general, the paper is well organized and written. The exposition in section 4 is well organized and relatively easy to follow. I appreciate that not all proofs are relegated to the appendix. Additionally, the notation preliminaries are complete and consistent with my expectations.
- **Novel theoretical contribution**: Theorem 1 (Eq. 8) provides an O(1/T) upper bound on the KL error that depends on the average mutual information between a token and the rest of the sequence. Importantly, the authors prove a matching lower bound, showing that their result is tight.
- **Significant improvements over prior work**: authors discuss the prior work of Feng et al. [6], and rightfully argue that their bound is more aligned with realistic values of L (sequence length) and n (vocabulary size).
- **Well-structured preliminaries**: section 1.2 carefully sets notation and clarifies the training objective, lowering the barrier to entry for readers outside the discrete diffusion community.
- **Sampling algorithm is accurate**: While some prior theoretical work might make unrealistic assumptions, this work assumes the true sampling algorithm of recent masked diffusion models such as MDLM [4].

---
## Weaknesses
**No empirical validation**: While the work is theoretical, a small-scale simulation on a synthetic distribution would strengthen the submission.

**Potentially large hidden constants**: Theorems quote unspecified "absolute constants" (C_1,C_2,C_3). The manuscript could be improved by including more details on what typical constants should be expected.

**Contextualization**: To better contextualize this work, the authors should consider citing a few recent works, which would make some claims in the introduction more precise. Specifically, by default, contrary to what is stated on lines 46-48, **sampling from masked diffusion models is generally slower than AR models**, despite parallel generation quality. Indeed, KV caching [1] accelerates AR models significantly, but is not available for models with bidirectional attention. However, diffusion language models can be accelerated through distillation [2,3]. For a comparison of the speed of text diffusion models with AR models, see Figure 2 of the appendix of [4]. Additionally, regarding the capabilities of diffusion language models, citing [5] would be relevant.


[1] Efficiently Scaling Transformer Inference, Pope et al.

[2] Beyond Autoregression: Fast LLMs via Self-Distillation Through Time, Deschenaux and Gulcehre

[3] Distillation of Discrete Diffusion through Dimensional Correlations, Hayakawa et al.

[4] Simple and Effective Masked Diffusion Language Models, Sahoo et al.

[5] Diffusion Language Models Can Perform Many Tasks with Scaling and Instruction-Finetuning, Ye et al.

[6] Theoretical Benefit and Limitation of Diffusion Language Model, Feng et al.

---

> ### Author Rebuttal · Authors · 2025-07-30
>
> We thank the reviewer for the insightful comments and thorough assessment of our work.
>
> **Weakness:**
> - Thank you for the suggestion. We add a synthetic numerical experiment in the revised manuscript to validate our theory.
>   - *Model.* we use a 1-D $K$-state Potts chain of length $L$ with coupling $J$ to model the distribution of text $X=(X^{(1)},\dots,X^{(L)})$. Concretely, $X^{(1)}\sim\text{Unif}([K])$ and $$\mathbb P\\{X^{(i)}=y|X^{(i-1)}=x\\}=\exp(J1\\{x=y\\})/(\exp(J)+K-1), \quad \forall x,y\in[K].$$ This construction allows us to compute explicitly the mutual information, the optimal mask predictor $p^\star(\cdot | X_t)$, and the densities of $p_{X_0}$ and $p_{Y_0|M}$.
>   - *Mask predictor.* We use the explicitly computable optimal mask predictor $p^\star(\cdot | X_t)$ to run the sampling process.
>   - *Sampling error.* Given the explicit densities of $p_{X_0}$ and $p_{Y_0|M}$, the expectation in the KL divergence, taken over both the mask schedule $M$ and the data distribution $p_{X_0}$, is approximated using Monte Carlo simulations.
>
>   - *KL divergence vs. mutual information.* For fixed $T=10$, $K=10$ and $L=100$, KL divergence vs. mutual information is summarized in the following table:
>
>     |   J   | Mutual Information |   KL   |
>     |:-----:|:------------------:|-------:|
>     | 0.000 |     4.80e-13       | 0.000  |
>     | 0.333 |     1.18e+00       | 0.051  |
>     | 0.667 |     5.53e+00       | 0.309  |
>     | 1.000 |     1.43e+01       | 0.748  |
>     | 1.333 |     2.86e+01       | 1.302  |
>
>     One can verify that the KL divergence error increases linearly with the mutual information.
>
>   - *KL divergence vs. number of iterations $T$.* With a balanced mask schedule, $K=10$, $L=100$ and $J=2$, $\log KL$ vs. $\log T$ is summarized in the following table.
>
>     | $T$ |   KL   | $\log$ KL | $\log T$ |
>     |-----|--------|-----------|-----------|
>     | 2   | 18.70  | 2.928     | 0.693     |
>     | 4   | 8.85   | 2.180     | 1.386     |
>     | 5   | 7.24   | 1.980     | 1.609     |
>     | 10  | 3.10   | 1.130     | 2.303     |
>     | 20  | 1.49   | 0.395     | 2.996     |
>     | 25  | 1.16   | 0.151     | 3.219     |
>     | 50  | 0.44   | –0.818    | 3.912     |
>
>     One can verify that the slope is very close to -1, demonstrating that the KL divergence error scales proportionally to $1/T$.
>
>   - *Conclusions.* Therefore, these two results validate our established sampling convergence theory.
>
>
> - Thank you for the suggestion.
> According to Theorem 1, one can see that $C_1 = T s_{\max}/\sum s_t \asymp 1$. In addition, a direct inspection of the proof of Theorem 2 shows that we may take $C_2 = \frac{1}{2}$ and $C_3 = \frac{1}{16}$. We will state these explicit constants in the revised manuscript.
>
> - Thank you for bringing these references to our attention and for the detailed description, which is very helpful for improving this paper. We’ll incorporate them in the revised manuscript to make the introduction more accurate.
>
> **Questions:**
> - This is certainly a valuable point! Classical results from information theory show that when the text distribution has a Markov structure, the mutual information between any token with the rest of the sequence is substantially smaller than that in the full dependent scenario. For instance, when $(X^{(1)}, X^{(2)}, \dots, X^{(L)})$ forms a first-order Markov chain, then $I(X^{(1)}; X^{(-1)})=I(X^{(1)}; X^{(2)})$, which is typically far less than that in the general case.
> Because our sampling bound depends on the same mutual-information terms, it automatically tightens for such structured data. Investigating diffusion language models under explicit structural assumptions (e.g., Markov, low-order dependencies) is therefore a promising avenue, and we will add a brief discussion of this point in Section 5.
>
> - Thank you for the suggestion. Our ongoing studies suggests that it is possible to extend our results to the entropy-based sampling and show its performance improvement over the random sampling. Specifically, in each iteration, we can rank the still-masked positions by their conditional entropy (which can be estimated using the learned mask predictor). Unmasking the lowest-entropy positions first reduces the per-step conditional mutual information (because mutual information is always upper bounded by entropy),  thereby leading to a smaller final error bound. Our conjecture is that when the entropy-based sampling procedure is used, the conditional mutual information terms in our current error bounds can be replaced by these smaller conditional entropy terms, which yields improved sampling accuracy. Providing a rigorous theoretical proof for this adaptive strategy is an interesting direction for future work.

---

> > ### Comment · Reviewer_v9hs · 2025-08-04
> >
> > Thank you for your thoughtful rebuttal. The synthetic experiment is relevant and interesting, and strengthen your submission. I have read all of your rebuttals, and vote to accept this paper, to be updated according to the points discussed in the rebuttal.

---

> > > ### Author Response · Authors · 2025-08-04
> > >
> > > Thank you for the encouraging feedback and for supporting acceptance!

---

### Official Review · Reviewer_zPJj · 2025-06-29

**Clarity:** 3
**Significance:** 3
**Originality:** 3
**Rating:** 5
**Confidence:** 1

**Summary:**

This paper investigates the theoretical understanding of generative *diffusion language models*. Unlike conventional autoregressive models, diffusion models allow parallel token sampling, which leads to faster generation and eliminates left-to-right generation constraints. The main results of the paper are convergences guarantees from an information-theoretic perspective: the sampling error, measured by KL divergence, decays inversely with the number of iterations and scales linearly with the mutual information between tokens in the target text sequence.  The paper establishes matching upper and lower bounds, which provides the tightness of their convergence analysis. These findings offer interesting theoretical insights into the practical effectiveness and robustness of diffusion language models, highlighting how the statistical dependencies in language data influence the efficiency of parallel diffusion sampling.

**Questions:**

My only question is about the generality of this statement:

*"Given accurate mask predictors, can we establish the convergence guarantees of diffusion language models
for general sampling procedures and data distribution?"*

In particular regarding the  *general sampling procedures and data distributions.*

Minor typos:

  - l6: diffusion models --> diffusion language models
  - l42: serves --> serve

**Ethical Concerns:**

["NO or VERY MINOR ethics concerns only"]

**Final Justification:**

This paper presents an interesting theoretical contribution by establishing convergence guarantees for diffusion language models using KL divergence and information-theoretic tools. The authors demonstrate the tightness and optimality of their analysis by providing matching upper and lower bounds, which reveal a fundamental relationship between sampling error, the number of diffusion steps, and the mutual information between tokens. While the theoretical proofs are complex, the work appears to be solid with no major weaknesses. The authors could, however, improve clarity by contextualizing the assumptions about general sampling procedures and data distributions earlier in the paper. Given its strong theoretical foundation and valuable contributions, in my opinion, the paper is a strong candidate for acceptance.

**Limitations:**

Limitations are correctly discussed.

**Paper Formatting Concerns:**

Paper is formatted correctly.

**Quality:**

3

**Strengths And Weaknesses:**

**Strengths:**

 - **Interesting theoretical contributions**: (As far as I know) the paper offers significant theoretical contributions by developing convergence guarantees for diffusion language models using KL divergence and IT tools. It quantifies the sampling error using KL divergence, showing an inverse decay with the number of steps  and a linear relationship with the mutual information between tokens in the target text sequence.

- **Tight Bounds**: The establishment of matching upper and lower bounds (up to constant factors) demonstrates the tightness and optimality of te convergence analysis, suggesting that the $1/T$ decay and mutual information dependence are fundamental limits.

- **Practical insights / intuition**: The results offer new insights into why diffusion language models are effective in practice.

- **Comparison with other relevant work:** The paper contrasts its findings with existing work, such as  the very recent work by Feng et al. 2025, showing a sharper and more general guarantee for arbitrary distributions, overcoming limitations of previous bounds that degrade for high-order n-gram models.

**Weaknesses:**

I don't see any major weakness in this work, though I recognize that I'm not able to correctly asses the technical proofs since this work is out of my domain of expertise. One point that could may be better discussed is the statement:

*"Given accurate mask predictors, can we establish the convergence guarantees of diffusion language models
for general sampling procedures and data distribution?"*

**General sampling procedures and data distributions.** The paper (as always) does make some assumptions about the sampling procedure and the data distribution, maybe this should be contextualized in the introduction. It would be nice to connect the assumptions to what we may observe on real data (there's some discussion about this at the very end).

---

> ### Author Rebuttal · Authors · 2025-07-30
>
> We are thankful for the reviewer’s valuable feedback and close attention to the manuscript.
>
> - For the question regarding generality, this indeed requires more clarification. First, our theory applies to any target text distributions---we impose no distributional assumptions whatsoever---so we refer to ``general data distributions.’’ Second, we analyze the **standard** sampling procedure of diffusion language models without introducing additional constraints or restrictions for analysis (e.g., we allow arbitrary mask sizes per iteration). Hence, we describe our results as covering “general sampling procedures” of diffusion language models.
>
> - Thank you for catching the typos. We have fixed them.

---

> > ### Comment · Reviewer_zPJj · 2025-08-04
> >
> > Thanks for addressing my comments. I'm happy to keep my acceptance score.

---

> > > ### Author Response · Authors · 2025-08-05
> > >
> > > Thanks again for your time and support!

---

### Official Review · Reviewer_DvD9 · 2025-07-03

**Clarity:** 2
**Significance:** 2
**Originality:** 3
**Rating:** 3
**Confidence:** 3

**Summary:**

This paper analyzes both the upper and lower bounds of the sampling error in diffusion language models from an information-theoretic perspective. It demonstrates that, under the assumption of linear dependence on the mutual information, the fundamental limit for the convergence rate of the sampling error is $O(1/T)$.

**Questions:**

Q1: What is the rationale for assuming that the sampling error has a linear dependence on the mutual information? Could the authors provide further justification or discussion for this assumption?

Q2: While the paper presents an analysis of sampling error convergence, could the authors offer any constructive suggestions on how to effectively reduce the sampling error or accelerate convergence for parallel sampling in diffusion language models?

Minor 1: The description in the Abstract (Line 6) is inaccurate: it should state “theoretical understandings of discrete diffusion language models” instead of “diffusion models”.

Minor 2: Why is reversal reasoning emphasized? This does not seem consistent with the nature of language generation, and I do not consider it an advantage.

Minor 3: The claim at Line 68 “This assumption does not align with practical diffusion language models that mask a large fraction of tokens at each iteration” needs supporting citations.

**Ethical Concerns:**

["NO or VERY MINOR ethics concerns only"]

**Final Justification:**

According to the rebuttal reply, I increased my rating by one point.

**Limitations:**

See Weaknesses and Questions.

**Quality:**

3

**Strengths And Weaknesses:**

## Strengths
Compared with previous results on diffusion models under n-gram distributions (e.g., Feng et al., 2025), this work is more general and rigorous, as it applies to arbitrary data distributions and token dependencies. The theoretical guarantees remain valid even for higher-order structures with long-range dependencies, unlike prior methods whose guarantees become loose or invalid as n increases.

## Weaknesses
While the paper provides a rigorous theoretical analysis, all derivations assume that the mask predictor can be trained to near-optimality, but no experimental results are provided to support the theoretical conclusions. In addition, the theoretical lower bound may not hold under certain extreme mask schedules (see Lines 283–290).

---

> ### Author Rebuttal · Authors · 2025-07-30
>
> We appreciate the reviewer’s time and effort in providing helpful feedback.
>
> **Weakness:**
> - This is certainly a valuable point that needs more clarification. We do **not** assume a near-optimal mask predictor. Our analysis only requires access to a pre-trained mask predictor (of arbitrary quality) and provides an off-the-shelf framework for assessing the text generation quality of diffusion language models. We decouple the mask predictor training phase from the sample generation stage, showing that the final error in KL divergence can be decomposed into (i) a training error term $\varepsilon_{train}$ that captures how far the learned predictor is from the optimal one, and (ii) a sampling error term incurred during the sample generation phase. The central contribution of this paper is to characterize the sampling error in terms of the iteration number and the statistical structure of the text distribution. Crucially, this sampling term is **independent** of the quality of the mask predictor. To make this point more precise and avoid misunderstanding, we will revise our statement about the pre-trained mask predictor from “We assume access to sufficiently accurate mask predictors” to “We assume access to some mask predictors”.
> - We add a synthetic numerical experiment in the revised manuscript to validate our theory.
>   - *Model.* we use a $K$-state Potts chain of length $L$ with coupling $J$ to model the distribution of text $X=(X^{(1)},\dots,X^{(L)})$. Concretely, $X^{(1)}\sim\text{Unif}([K])$ and $$\mathbb P\\{X^{(i)}=y|X^{(i-1)}=x\\}=\exp(J1\\{x=y\\})/(\exp(J)+K-1), \quad \forall x,y\in[K].$$ This construction allows us to compute explicitly the mutual information, the optimal mask predictor $p^\star(\cdot | X_t)$, and the densities of $p_{X_0}$ and $p_{Y_0|M}$.
>   - *Mask predictor.* We use the explicitly computable optimal mask predictor $p^\star(\cdot | X_t)$ to run the sampling process.
>   - *Sampling error.* Given the explicit densities of $p_{X_0}$ and $p_{Y_0|M}$, the expectation in the KL divergence, taken over both the mask schedule $M$ and the data distribution $p_{X_0}$, is approximated using Monte Carlo simulations.
>
>   - *KL divergence vs. mutual information.* For fixed $T=10$, $K=10$ and $L=100$, KL divergence vs. mutual information is summarized in the following table:
>
>     |   J   | Mutual Information |   KL   |
>     |:-----:|:------------------:|-------:|
>     | 0.000 |     4.80e-13       | 0.000  |
>     | 0.333 |     1.18e+00       | 0.051  |
>     | 0.667 |     5.53e+00       | 0.309  |
>     | 1.000 |     1.43e+01       | 0.748  |
>     | 1.333 |     2.86e+01       | 1.302  |
>
>     One can verify that the KL divergence error increases linearly with the mutual information.
>
>   - *KL divergence vs. number of iterations $T$.* With a balanced mask schedule, $K=10$, $L=100$ and $J=2$, $\log KL$ vs. $\log T$ is summarized in the following table.
>
>     | $T$ |   KL   | $\log$ KL | $\log T$ |
>     |-----|--------|-----------|-----------|
>     | 2   | 18.70  | 2.928     | 0.693     |
>     | 4   | 8.85   | 2.180     | 1.386     |
>     | 5   | 7.24   | 1.980     | 1.609     |
>     | 10  | 3.10   | 1.130     | 2.303     |
>     | 20  | 1.49   | 0.395     | 2.996     |
>     | 25  | 1.16   | 0.151     | 3.219     |
>     | 50  | 0.44   | –0.818    | 3.912     |
>
>     One can verify that the slope is very close to -1, demonstrating that the KL divergence error scales proportionally to $1/T$.
>
>   - *Conclusions.* Therefore, these two results validate our established sampling convergence theory.
>
>   In addition, because the true data distribution is unknown in the real-world datasets, it is infeasible to compute the token-level mutual information or the KL divergence between the output and the data distribution. Hence, we restrict our numerical experiments to the synthetic data.
> - Our lower bound targets the balanced mask schedules used in practice, where the number of unmasked tokens per iteration is of the same order. Admittedly, the bound can fail for deliberately unbalanced mask schedules---for instance, unmasking half the sequence in the first iteration and then proceeding token-by-token in the rest of iterations. Because balanced mask schedules are much more commonly employed in practice, we believe this exclusion does not compromise the practical insights of our results.
>
> **Questions:**
> - “The sampling error has a linear dependence on the mutual information” is the main result of this paper (upper bounds in Theorem 1 and Corollary 1, and lower bounds in Theorem 2), rather than an assumption.
> - Thank you for the suggestion. A practical implication of our theory is to unmask, at each iteration, the tokens whose conditional dependence on the rest of the sequence is weakest. Concretely, Eq. (24) shows that the per-step contribution to the total sampling error is the conditional mutual information between a newly revealed token and the remaining tokens. Hence, selecting positions with smaller conditional mutual information minimizes that contribution. Moreover, because $I(X;Y|Z)\leq H(X|Z)$ for any random variables $X,Y,Z$, we can implement this strategy by ranking tokens by conditional entropy: at step $t$, use the learned mask predictor $\hat{p}_i(\cdot | Y_t)$ to estimate the conditional entropy of the token at position $i$. Unmasking the positions with the lowest conditional entropy provides a simple heuristic that approximates the mutual-information criterion without additional training or external estimates. We will add a remark on this point in the revised manuscript.
> - The original “diffusion model approaches” means using diffusion models for language models. We have changed it to “diffusion language models” to avoid this potential misunderstanding.
> - Reversal reasoning is one key practical motivation of diffusion language models (see e.g., [1]). This task spans many real-world scenarios, including cloze/fill-in-the-blank problems, and span infilling. For example, literature examples often ask students to recover the first half of a poem from the second half; language exams routinely require predicting a verb given its object.
> - Thank you for the suggestion. We have added the reference [2] to show that practical diffusion language models indeed mask more than one token in average at each iteration.
>
> [1] Nie, S., Zhu, F., You, Z., Zhang, X., Ou, J., Hu, J., Zhou, J., Lin, Y., Wen, J.-R., and Li, C. (2025).
> Large language diffusion models. arXiv preprint arXiv:2502.09992.
>
> [2] Yu, R., Li, Q., & Wang, X. (2025). Discrete Diffusion in Large Language and Multimodal Models: A Survey. arXiv preprint arXiv:2506.13759.

---

> > ### Comment · Reviewer_DvD9 · 2025-08-04
> >
> > Thank you for the detailed response. I find the use of information-theoretic perspectives to explain the convergence behavior of Diffusion Language Models both insightful and compelling. I’ve increased my rating by one point, and I hope the authors will consider including the synthetic numerical experiment in the final version.
> >
> > That said, I did not provide a more positive rating because, while the response proposes an interesting optimization heuristic—“Unmasking the positions with the lowest conditional entropy provides a simple heuristic that approximates the mutual-information criterion without additional training or external estimates”—it lacks preliminary experimental evidence to support this claim. Therefore, this weakness remains.

---

> > > ### Author Response · Authors · 2025-08-04
> > >
> > > Thank you for your thoughtful feedback and the improved rating. All concerns except Q2 have been addressed in the revision.
> > >
> > > Our focus and contribution is primarily theoretical: we establish theoretical guarantees on the sampling error of diffusion language models. We now add a remark on the “low-conditional-entropy” unmasking schedule inspired by our theory; however, a comprehensive empirical evaluation lies beyond the scope of this paper and is left for future work.

---

### Official Review · Reviewer_hh8p · 2025-07-04

**Clarity:** 3
**Significance:** 2
**Originality:** 2
**Rating:** 4
**Confidence:** 3

**Summary:**

Masked diffusion language models (DLMs) generate text by iteratively unmasking subsets of tokens in parallel, potentially sampling much faster than autoregressive transformers. Despite strong empirical results, there has been little theory quantifying how many unmasking iterations $T$ are needed to approach the true data distribution. This paper fills that gap by analyzing the sampling phase—assuming an already-trained, sufficiently accurate mask-prediction network—and derives information-theoretic bounds on the KL divergence between model output and data.

In the main theorems, the authors proved an explicit upper bound on the KL divergence between the model output and the data distribution that decays as $O(1/T)$ with the number of unmasking iterations $T$ (for balanced schedules) and depends linearly on a sequence-level mutual-information term. This bound is also shown to be unimprovable under the paper’s oracle-predictor assumption.

**Questions:**

Please refer to the Weaknesses Section above.

**Ethical Concerns:**

["NO or VERY MINOR ethics concerns only"]

**Final Justification:**

The authors added a synthetic numerical experiment in the updated manuscript to validate their theory, which also answered most of my questions on experiments.

**Limitations:**

There is no negative societal impact of their work,

**Paper Formatting Concerns:**

There is no paper formatting concerns in this work.

**Quality:**

2

**Strengths And Weaknesses:**

Strengths:

The paper makes a compelling theoretical contribution by establishing the first rigorous relationship between the number of unmasking iterations and generation quality for masked-diffusion language models.  It offers a tight information-theoretic analysis with matching upper- and lower-bound rates, while scaling gracefully to schedules that update many tokens in parallel.  A particularly valuable aspect is its identification of sequence-level mutual information as the key statistic governing convergence, which provides an intuitive bridge between linguistic dependence and iteration complexity.  The authors further separate sampling-phase error from training-phase error, clarifying where future architectural or data improvements are most beneficial.  Beyond its theory, the manuscript is clearly written, logically structured, and supported by a thorough discussion of prior work, making the results accessible to a broad audience.

Weaknesses and Questions:
1. For the oracle predictor assumption, the main bound becomes vacuous when $\varepsilon_{\text{train}}$ dominates; no scaling law links this term to model capacity or data. It would be better to add some discussions on the $\varepsilon_{\text{train}}$ term as well as its connection with model capacity.
2. Is there any synthetic or real experiments that the authors can provide to confirm the predicted inverse-linear curve or the practical relevance of the mutual-information coefficient?
3. For the potential gap in KL decomposition, I wonder whether there is a $\varepsilon_{\text{train}}$ term omitted at every step when the predictors are imperfect, which will loosen the final bound in this case. I think the authors should add a remark after Lemma 1, or make the Lemma 1 more robust.
4. I wonder if the self-conditioning breaks the independence assumption used in Lemma 1, and how would that affect the bound?

---

> ### Author Rebuttal · Authors · 2025-07-30
>
> We appreciate the reviewer’s thoughtful comments and careful evaluation.
>
> - Thank you for the suggestion. We agree that it would be beneficial to add an explicit remark on $\varepsilon_{train}$, which reads as follows. Our theory shows that the overall error in KL divergence exhibits a two-stage behavior: when the number of iterations $T<C_1\sum_i I(X^{(i)};X^{(-i)})/\varepsilon_{train}$ where $C_1$ is the constant from Corollary 1, the sampling error term dominates and the overall error decays proportionally to $1/T$; once $T$ exceeds this threshold, the error plateaus and becomes almost insensitive to further increases in $T$.
> Deriving a sharp scaling law for $\varepsilon_{\text{train}}$ itself would require new techniques to analyze the model capacity of transformers and the sample complexity of their nonconvex training, which is currently unavailable and beyond the scope of the present work.
>
> - Thank you for the suggestion. We add a synthetic numerical experiment in the revised manuscript to validate our theory.
>   - *Model.* we use a $K$-state Potts chain of length $L$ with coupling $J$ to model the distribution of text $X=(X^{(1)},\dots,X^{(L)})$. Concretely, $X^{(1)}\sim\text{Unif}([K])$ and $$\mathbb P\\{X^{(i)}=y|X^{(i-1)}=x\\}=\exp(J1\\{x=y\\})/(\exp(J)+K-1), \quad \forall x,y\in[K].$$ This construction allows us to compute explicitly the mutual information, the optimal mask predictor $p^\star(\cdot | X_t)$, and the densities of $p_{X_0}$ and $p_{Y_0|M}$.
>   - *Mask predictor.* We use the explicitly computable optimal mask predictor $p^\star(\cdot | X_t)$ to run the sampling process.
>   - *Sampling error.* Given the explicit densities of $p_{X_0}$ and $p_{Y_0|M}$, the expectation in the KL divergence, taken over both the mask schedule $M$ and the data distribution $p_{X_0}$, is approximated using Monte Carlo simulations.
>
>   - *KL divergence vs. mutual information.* For fixed $T=10$, $K=10$ and $L=100$, KL divergence vs. mutual information is summarized in the following table:
>
>     |   J   | Mutual Information |   KL   |
>     |:-----:|:------------------:|-------:|
>     | 0.000 |     4.80e-13       | 0.000  |
>     | 0.333 |     1.18e+00       | 0.051  |
>     | 0.667 |     5.53e+00       | 0.309  |
>     | 1.000 |     1.43e+01       | 0.748  |
>     | 1.333 |     2.86e+01       | 1.302  |
>
>     One can verify that the KL divergence error increases linearly with the mutual information.
>
>   - *KL divergence vs. number of iterations $T$.* With a balanced mask schedule, $K=10$, $L=100$ and $J=2$, $\log KL$ vs. $\log T$ is summarized in the following table.
>
>     | $T$ |   KL   | $\log$ KL | $\log T$ |
>     |-----|--------|-----------|-----------|
>     | 2   | 18.70  | 2.928     | 0.693     |
>     | 4   | 8.85   | 2.180     | 1.386     |
>     | 5   | 7.24   | 1.980     | 1.609     |
>     | 10  | 3.10   | 1.130     | 2.303     |
>     | 20  | 1.49   | 0.395     | 2.996     |
>     | 25  | 1.16   | 0.151     | 3.219     |
>     | 50  | 0.44   | –0.818    | 3.912     |
>
>     One can verify that the slope is very close to -1, demonstrating that the KL divergence error scales proportionally to $1/T$.
>
>   - *Conclusions.* Therefore, these two results validate our established sampling convergence theory.
>
>   In addition, because the true data distribution is unknown in the real-world datasets, it is infeasible to compute the token-level mutual information or the KL divergence between the output and the data distribution. Hence, we restrict our numerical experiments to the synthetic data.
>
> - We appreciate this careful point. Lemma 1 is **not** ``missing” $\varepsilon_{train}$: the term $\varepsilon_{train}$ is fundamentally absent in Lemma 1 because this lemma analyzes the sampling error when the optimal mark predictor $p^\star$ is used. Error due to imperfect training is aggregated into $\varepsilon_{train}$ and handled in (17), which relates the distribution induced by the learned predictor $\hat{p}$ to that induced by $p^\star$. This separation decouples training from sampling, which allows Lemma 1 to focus on  $KL(p_{X_0}\|p_{Y_0^\star|M})$. We will add a remark after Lemma 1 to make this point clearer.
>
> - In the training stage, the mask predictor is parametrized such that tokens in the unmask set are predicted independently given the current context (see (4) and (5)). Therefore, the proof of Lemma 1 reflects this design: the gap between the factorized predictor and the true joint conditional over a block of tokens is controlled by conditional mutual information terms that precisely measure the statistical dependence among those tokens after conditioning on the revealed ones. This is why conditional mutual information naturally appears in the per-step error decomposition.

---

### Official Review · Reviewer_2wt9 · 2025-07-23

**Clarity:** 4
**Significance:** 4
**Originality:** 4
**Rating:** 5
**Confidence:** 4

**Summary:**

The paper sets out to prove an upper and lower bound on the error of the sampling distribution of a trained masked diffusion model relative to the true sequence distribution it is trained on. The authors bound the expected KL divergence (over choices of mask sets at sampling) above by the training error as well as the mutual information of each token and the rest of the sequence with a 1/L prefactor, where L is the length of the sequence. The authors then refine the bound and show that this 1/L prefactor holds in both the upper and lower bounds with a similar weighted mutual information sum of each token and chunks of the remaining sequence, showing the sampling error bound is tight.

**Questions:**

- Are there insights from your theoretical analysis that could illuminate if it is possible to deliberately choose masking sets (or an unmasking ordering based on mutual information) at sampling time to reduce the sampling error from the true distribution?

**Ethical Concerns:**

["NO or VERY MINOR ethics concerns only"]

**Final Justification:**

- The paper is strong and insightful towards understanding theoretical limitations of sampling with masked diffusion models. Responses from the authors during rebuttals confirm the original positive impression.

**Limitations:**

The authors have adequately addressed the limitations.

**Quality:**

4

**Strengths And Weaknesses:**

Strengths:
- The authors examine a very fundamental theoretical question regarding how far the sampling distribution of masked diffusion models is from the true sequence distribution, and offer a comprehensive answer.

- The authors identify that beyond training error, there is a fundamental error stemming from the product posterior assumption.

- The exposition is extremely well done.

Weaknesses:
- The proofs in the appendix could be made even more clear with a simple "English" summary of some of the equations. For example, in (22), some brief sentence right after like "We break down the the KL divergence between the "ideal" sampling distribution and the sequence distribution into a sum of expected KL divergences between the true and productized marginals at one "unmasking" sampling step" (or something along these lines). Essentially reminding the reader how $D_t$ and similar variables are defined w.r.t. the unmasking sampling process.

---

> ### Author Rebuttal · Authors · 2025-07-30
>
> We thank the reviewer for the careful reading and the constructive feedback.
>
> **Weakness:**
> - Thank you for the suggestions. We will add plain-language explanations in the proofs. For instance, we will add the suggested summary for the KL divergence in (22); we will discuss the intuition for the KL divergence decomposition in (23); we will include an explanation for the conditional mutual information in (24).
> To aid navigation, we will also (i) restate the sampling update (6) and the factorized predictor (4), and (ii) remind readers of the masking sets $W_t$ and $D_t$ throughout the proof.
>
> **Questions:**
> - We appreciate this thoughtful point. A practical implication of our theory is to unmask, at each iteration, the tokens whose conditional dependence on the rest of the sequence is weakest. Concretely, (24) shows that the per-step contribution to the total sampling error is the conditional mutual information between a newly revealed token and the remaining tokens. Hence, selecting positions with smaller conditional mutual information minimizes that contribution. Moreover, because $I(X;Y|Z)\leq H(X|Z)$ for any random variables $X,Y,Z$, we can implement this strategy by ranking tokens by conditional entropy: at step $t$, use the learned mask predictor $\hat{p}_i(\cdot | Y_t)$ to estimate the conditional entropy of the token at position $i$. Unmasking the positions with the lowest conditional entropy provides a simple heuristic that approximates the mutual-information criterion without additional training or external estimates. We will add a remark on this point in the revised manuscript.

---

> > ### Comment · Reviewer_2wt9 · 2025-08-03
> >
> > I thank the authors for their thoughtful responses and maintain my original positive score.

---

> > > ### Author Response · Authors · 2025-08-04
> > >
> > > Thank you again for your careful review and support!

---

### Decision · Program_Chairs · 2025-09-17

**Decision:**

Accept (poster)

**Comment:**

This paper establishes theoretical convergence guarantees for discrete diffusion language models, proving that sampling error (measured by KL divergence) decays as O(1/T) with the number of denoising iterations T, where the constant factor depends linearly on mutual information between tokens in the target sequence. The authors provide matching upper and lower bounds, demonstrating the tightness of their analysis. Strengths include the novel theoretical contribution addressing a fundamental gap in understanding diffusion language models, rigorous information-theoretic analysis with tight bounds, excellent exposition and clarity, and significant improvements over prior work that was limited to n-gram distributions. Weaknesses center on the lack of empirical validation, assumptions about mask predictor quality that may not reflect practice, and potential concerns about hidden constants in the bounds.

The author rebuttal successfully addressed most reviewer concerns, leading to improved scores from multiple reviewers. Key additions included synthetic numerical experiments validating the theoretical predictions (addressing concerns from 2wt9, hh8p, DvD9, and v9hs), clarification that the analysis doesn't assume near-optimal mask predictors but works with arbitrary quality predictors (addressing DvD9's confusion), and explicit constants for the bounds (addressing v9hs's concern). Reviewer hh8p upgraded from "borderline accept" to accept after seeing the synthetic experiments, while DvD9 increased their rating by one point. However, DvD9 maintained a remaining concern about the proposed optimization heuristic (unmasking tokens with lowest conditional entropy) lacking experimental validation, noting this as a continuing weakness despite the overall positive theoretical contribution. The synthetic experiments and clarifications were sufficient to demonstrate the paper's theoretical validity and practical relevance, with the remaining empirical gap being acceptable.